# Additive manufacturing of multi-material and hollow structures by Embedded Extrusion-Volumetric Printing

Silvio Tisato [1], Grace Vera[1], Qingchuan Song[2,3], Niloofar Nekoonam[1,2] & Dorothea Helmer [1,2,3] ✉

Tomographic volumetric additive manufacturing support-free 3D printing has significantly faster print speed than traditional vat photopolymerization and material extrusion techniques. At the same time, tomographic volumetric additive manufacturing allows the embedding of external objects in the print volume before the print to produce complex multi part assemblies by so-called overprinting. As tomographic volumetric additive manufacturing increases its popularity, more and more of its limitations with regards to available materials are removed, and printing with acrylates, ceramics and glasses has been shown. However, creating multi material parts without resorting to overprinting is still challenging. Here, multi-material tomographic volumetric additive manufacturing printing is presented by combining tomographic volumetric additive manufacturing and Embedded 3D printing: photopolymerizable support baths are used to deposit inks in defined shapes allowing the consecutive definition and volumetric printing of multi-material structures via tomographic volumetric additive manufacturing. We show the fabrication of several multi-material structures with arbitrarily positioned materials, as well as chips with microchannels having diameters lower than 120 μm.

The introduction of additive manufacturing represented a paradigm shift in the way that prototyping and production of complex geometry is carried out, allowing to create shapes not possible with traditional subtractive manufacturing techniques. Since its introduction, many different techniques have been shown, each with their own set of advantages and disadvantages. The most common 3D printing methods are based on material extrusion or vat photopolymerization, building the 3D part by completing 2D layers and stacking them, usually in the z-direction. In material extrusion methods, the desired part is built by depositing materials following a 3D path—such as thermoplastics in Fused Deposition Modeling (FDM) and pastes in Direct Ink Writing (DIW)[1,2]. This allows to produce complex geometries with minimal material waste. In vat photopolymerization-based

methods, a vat containing a photopolymerizable resin is exposed to stimuli such as UV light that cause the resin to polymerize, producing the desired part. Examples of vat photopolymerization include Stereolithography (SLA), where a laser beam is scanned on the resin, locally polymerizing it, and MSLA (Masked Stereolithography), where a light beam is masked to expose the desired 2D projection onto the resin, polymerizing a 2D layer at once[3]. While 3D printing is very promising, it still faces some drawbacks. Material extrusion methods are generally slow, as they typically build up the 3D structure by extruding lines, requiring moving the extrusion head to each point of the desired 3D object[4]. Also, the method produces considerable material waste when printing complex parts that require the use of support structures[5]. At the same time, material extrusion easily allows to

[1]Freiburg Materials Research Center (FMF), University of Freiburg, Stefan-Meier-Str. 21, 79104 Freiburg, im Breisgau, Germany. [2]IMTEK - Laboratory of Process Technology, University of Freiburg, Georges-Köhler-Allee 103, 79110 Freiburg, im Breisgau, Germany. [3]Freiburg Center of Interactive Materials and Bioinspired Technologies (FIT), University of Freiburg, Georges-Köhler-Allee 105, 79110 Freiburg, Germany. ✉e-mail: dorothea.helmer@imtek.uni-freiburg.de

integrate multiple materials in a single printed object by switching the material being extruded or using multiple print heads[6]. Vat photo-polymerization methods show higher resolution and are generally faster than material extrusion, as they can use galvanometers to achieve high scan speeds or simply polymerize a 2D layer at once[7,8]. However, they do not easily allow to produce multi-material objects as they typically use low viscosity photopolymerizable resins, which do not allow placing multiple resins in a single vat without mixing or require vat switching multiple times per layer to produce multi-material structures.

Tomographic volumetric additive manufacturing (VAM or TVAM) is a vat photopolymerization technique that has been recently introduced[9,10]. It enables rapid printing of 3D structures by exposing the full structure all at once as 2D projections that propagate through a rotating resin vial, drastically reducing print times when compared to traditional 3D printing techniques[11]. The printer setup for VAM generally needs to provide a quasi-collimated beam for the 2D projections that are imaged into the vial, which typically leads to a high complexity of the optical system, although there have been efforts to remove this assumption or engineer simpler systems[12–16]. The assumption of collimated light propagation allows to simplify the reconstruction process and to successfully reproduce the desired 3D dose distribution. VAM requires the resin to be transparent to the wavelength used in the printing process, typically between 385 nm and 525 nm, which limits the types of materials that can be used. So far, acrylates, thiols, hydrogels, ceramics, and glass have been shown to work for the VAM process[10,17–21]. VAM resins also generally require a rather high viscosity to reduce sinkage effects from the densification of the part during the printing, with usual VAM resins having viscosities above 2000 mPa s[22]. To avoid issues with the part sinking during the print, volumetric printing can also be executed in a gel or solid where the non-polymerized resin is dissolved and washed out after the print is complete[23–25].

However, multi-material 3D printing is still a challenge in VAM. Current methods for generating multi-material prints in TVAM include integrating a previously fabricated structure into a TVAM vial and curing around the structure[26], generating stiffness gradient materials by selectively curing an acrylate-epoxy mixture in a dual-wavelength setup[27], and varying material in the vertical direction by stacking different material layers[28]. Another method is sequential multi-material volumetric printing (SMVP), where an initial structure is printed in a first material, then the excess resin is washed out and the vial is filled with a second material which is then used to perform volumetric printing of the second part of the structure, similar to a vat-exchange process[28,29]. These methods significantly broadened the capabilities of volumetric printing, but require complex alignment steps after resin exchange and additional support structures or are limited to mixtures of materials with orthogonal curing wavelengths requiring multi-wavelength volumetric printers. A promising method for multi-material VAM is the combination of embedded 3D printing (EMB3D) and TVAM, which is termed Embedded Extrusion-Volumetric Printing (EmVP)[30]. EMB3D is a material extrusion technique that deposits an ink into a supporting bath using a needle[31]. The supporting bath must possess specific rheological properties that allow the needle to traverse the support bath while at the same time keeping the deposited liquid ink in place until the completion of the print[32]. The printed parts can be polymerized all at once when using polymerizable inks in the additive EMB3D process[33]. Alternatively, in the subtractive EMB3D process, the deposited ink is a sacrificial ink, while the support bath can be polymerized: once the bath is cured, the ink can be removed, which has been shown to produce microfluidics channels[34,35]. Additive EMB3D, being a material extrusion process, can be easily used to produce multi-material structures, but requires long print times when producing complex structures, and can suffer from print accuracy issues due to the interaction between ink and support bath[32,36,37]. EmVP

has previously been used by Ribezzi et al. and Riffe et al. to bioprint multi-material VAM parts by embedding secondary material areas in the resin and overprinting bulk structures around them[29,30,38]. Taken together, all previously reported EmVP methods show overprinting, thus including one material—the ink of the embedded print or another object—in another material that is then polymerized by TVAM. None of the previously reported methods were able to produce multi-material structures in a single volumetric print step where different features of the structures are composed of different materials along the horizontal and vertical direction, without the need for support structures or vat exchange.

In this work, we show an EmVP process capable of precisely depositing an ink into a photopolymerizable supporting bath through EMB3D printing, followed by rapid VAM curing of the combined structure. When the embedded ink is photopolymerizable and designed to cure at similar times as the support bath, this results in fast printing of multi-material parts all with exposed, free standing surfaces by simultaneous volumetric curing of both materials thanks to the matched curing times, while when the embedded ink is not photo-polymerizable we obtain 3D parts with precise embedded freeform microchannels. We present the development of the photopolymerizable supports baths and inks as well as the characterization of the printed multi-material parts and microchannels, showing several multi-material structures as well as microchannels with diameters down to less than 120 μm.

## Results

### Positive Embedded Extrusion-Volumetric Printing

By combining the support baths and deposition process typical of EMB3D and the high-speed volumetric curing process of VAM, we obtained a print process that can easily produce multi material structures as well as structures with 3D embedded channels. The main steps of the process, which we term positive Embedded Extrusion-Volumetric (EmVP) are shown in Fig. 1. First, a vial of polymerizable support bath is loaded onto a custom EMB3D printer and the selected ink is embedded into the volume. Then, the vial is moved to a custom VAM printer previously developed by our group and the desired shape is cured via VAM, thus producing a part with embedded ink. In the case where this embedded ink is photocurable (Fig. 1a–d), both ink and support bath cure together, resulting in a multi-material part. Here we show the formation of parts with soft and hard properties, with the soft material having an elastic modulus of 1.28 and shore hardness of 4.9 and the hard material having an elastic modulus of 122 MPa and shore hardness of 27.4 (see Fig. 1e and Supplementary Fig. 10). To achieve the formation of a part, the gelation time of both materials must be similar (see Fig. 1f). In the case of an embedded ink that does not cure during the VAM process, we obtained a negative EmVP part which produces embedded hollow channels, Fig. 1g–h. For the prints, we used two distinctive deposition strategies for the embedded material. We used either targeted deposition of the embedded ink, Fig. 1i, which only deposits the minimum amount of material closely reproducing the desired embedding region, or area deposition of the embedded ink, Fig. 1j, which deposits ink in a wider region than necessary and relies completely on the VAM step to provide the final structure. These methods can help to balance EMB3D print time and resolution of the final structures, as EMB3D resolution changes with extruded volume and print speed. For the same amount of embedded material, a higher resolution requires a higher EMB3D print time, while with VAM, the print time is mainly controlled by how fast the resin can be pushed past the gelation point.

### Positive EmVP

To showcase the ability of the positive EmVP process to produce multi material structures, we have selected two material with different mechanical properties. The first one is a diacrylate (HDDA),

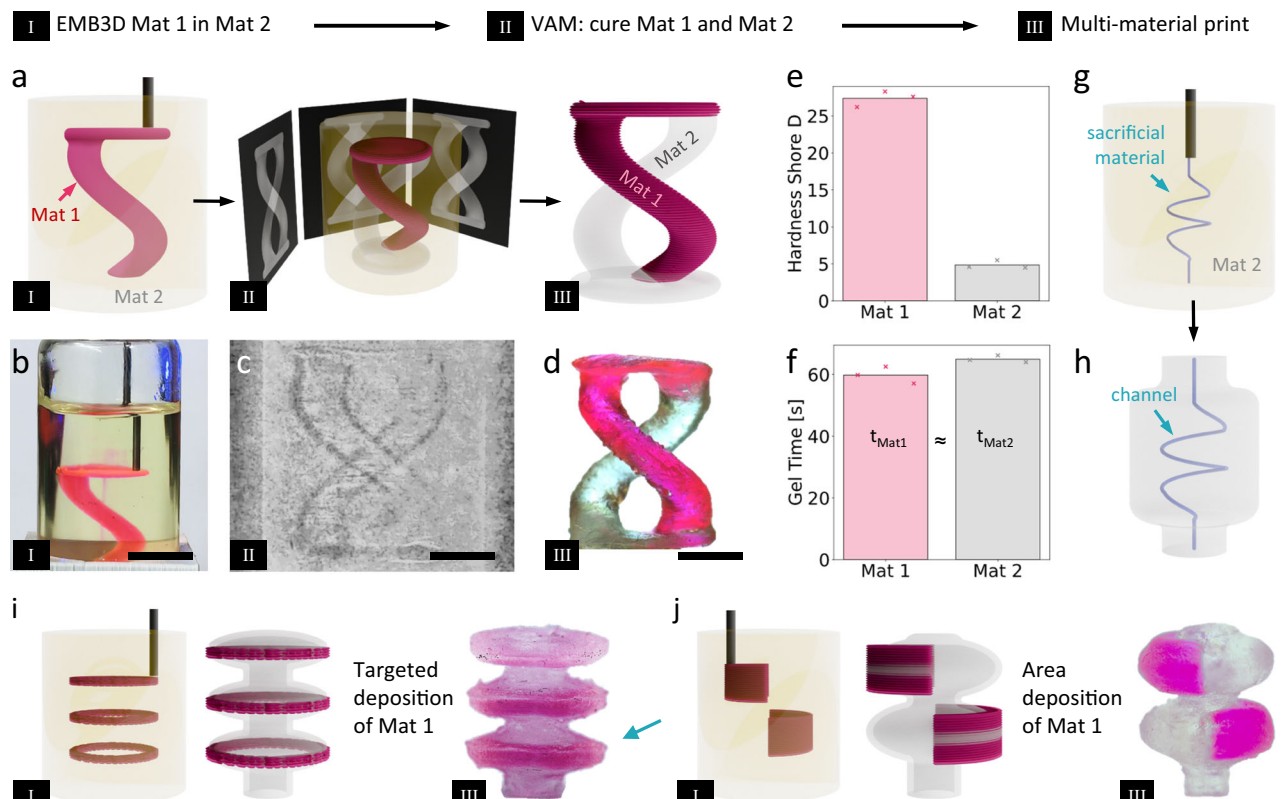

**Fig. 1 | Schematic representation of the positive and negative Embedded Extrusion-Volumetric Printing (EmVP) print processes using supporting baths.** **a** The desired embedded material, Mat 1, is deposited through EMB3D in a vial filled with photopolymerizable supporting bath, Mat 2. After embedding of the ink, VAM of the desired part is carried out, where the two liquid materials are cured into the desired final shape. When both embedded material and support bath are polymerizable, this results in a multi-material part through positive EmVP process, here a two-material helix is shown. **b** Shows the deposition of the ink, (**c**) shows the formation of the structure with VAM, (**d**) shows the final two-material helix. **e** The materials designed here have very different Shore D hardness of 27.4 and 4.9,

respectively. **f** The gelation times were engineered to be close to enable the two materials to cure together into a single structure. **g** In the negative EmVP process, the ink being deposited is a non-polymerizable sacrificial material which is deposited in the shape of the desired microchannel. **h** After VAM printing of the surrounding structure, the sacrificial material is flushed out, resulting in a part with embedded microchannels. **i** In the positive EmVP process, the embedding of material can be targeted—thus creating the distinct structure of the part by EMB3D. **j** Alternatively, the deposition can happen in a wider area, relying on the VAM step to provide the desired resolution in the final structure. Scale bars: (**b**) 9 mm, (**c, d**) 4.5 mm, (**i**) 5 mm, (**j**) 5.5 mm.

characterized by a high elastic modulus of 122 MPa and a shore D hardness of 27.4. The second one is an aliphatic urethane acrylate, Genomer 1122TF (G1122TF), with an elastic modulus of 1.28 MPa and a shore D hardness of 4.9. Both of these materials are not suitable for VAM as-is, as they have a low viscosity of 50 mPa s and 10 mPa s for G1122TF and HDDA, respectively, which would cause the partially polymerized parts to sediment during printing. To enable the EmVP process, we have engineered the materials to form photopolymerizable support baths by adding Aerosil R805 as a rheology modifier. The hydrogen bonds between the silica form a network that increases the apparent viscosity of the material and at certain concentrations forms a gel with thixotropic properties, allowing the deposition needle to travel through it while at the same time retaining the embedded ink at the deposited position. In our case, an addition of 7 wt% of R805 to HDDA and 8 wt% of R805 in G1122TF provide suitable properties to act as an embeddable ink in the case of HDDA (Mat 1) and as a support bath in the case of G1122TF (Mat 2), both of which are suitable for VAM. Differences in refractive index of the materials deviate the light propagation direction from the assumed straight line. To reduce the effect of refractive index mismatch, we have chosen materials that show close refractive indices, as reported by the suppliers, with 1.4598 and 1.456 for Mat 1 and Mat 2, respectively. To demonstrate the capabilities of positive EmVP to produce complex, multi material structures, we have printed several structures as shown in Fig. 2. In Fig. 2a we show a rendition of the famous Thinker by Rodin, where the stone is printed

with the stiffer HDDA material. The positive EmVP process can produce stacked multi-material structures, by filling the print volume with layers of different material, as shown in Fig. 2b,c, where a simple skeleton sphere structure is printed half in Mat 1 (bottom side) and half in Mat 2 (top side). Note that the embedded Mat 1 does not need to closely follow the shape of the final structure thanks to curing only selected areas inside the deposited Mat 1. This allows us to reduce the impact of common EMB3D issues such as unwanted oozing of ink during travel moves, shown in Supplementary Fig. 7a,b, by only polymerizing selected regions of the deposited ink, as shown in Supplementary Fig. 1. A close up of the structure in Fig. 2b is included in Fig. 2d, showing the transition between the two materials.As shown Fig. 2b, c, when a weight is placed on the structure, only the less stiff Mat 2 side buckles, while the Mat 1side is unaffected, following from the difference in hardness shown in Fig. 1d. A video of the structure being compressed is provided in Supplementary Movie 1. Additionally, the positive EmVP process can embed selected materials in specific areas of the structure, as shown in Fig. 2e, g, k. This is particularly useful to generate anisotropies in the printed parts, as in Fig. 2g, where we have printed a lattice structure where each of the two diagonal halves are printed in one of the two materials. A magnified view is shown in Fig. 2h highlighting the transition between the different materials. By calculating the Hausdorff distance, i.e. the average distance between the surface of the printed multi-material structure and that of the 3D model, shown in Supplementary Fig. 8, we obtain an

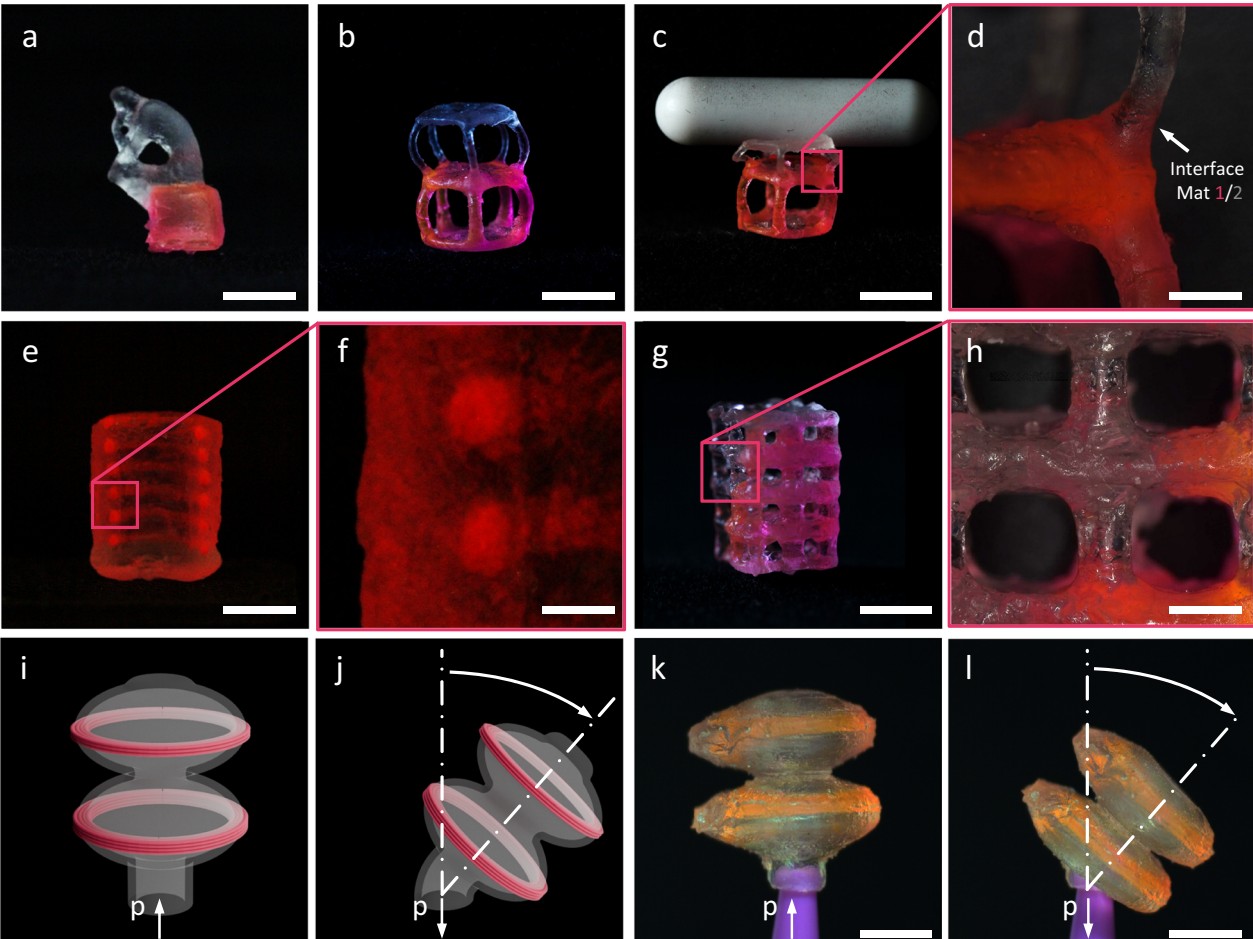

**Fig. 2 | Selection of parts printed with the positive EmVP process. a** The famous Thinker by Rodin was printed in Mat 2 while using the stiff Mat 1 embedded for targeted deposition for the stone on which it sits. **b** Skeleton sphere printed by depositing the shell of the lower side in Mat 1 by area deposition and curing only the desired features. **c** The structure from (**b**) with a magnetic stirring bar weighing 3 g placed on top, showing the difference in hardness between Mat 1 and Mat 2. **d** Close up of the structure shown in (**b**) showing the interface between Mat 1 and Mat 2. **e** Cross sectional cut of a hollow cylinder in Mat 2 with Mat 1 deposited by targeted embedding of a filament. **f** Close up of the filaments cross sections in (**e**), showing filament diameter in the range of 750 μm. **g** Lattice structure with Mat 1 embedded in Mat 2 by area deposition, splitting the structure diagonally in two between Mat 1 and Mat 2. **h** Close up view of the holes in the lattice side, showing print resolution and gradual transition following between Mat 1 and Mat 2 following the diagonal. **i, k** 3D Model and printed part of a hollow bellow with embedded reinforcement rings serving as a pneumatic actuator, with reinforcement rings in Mat 1 deposited by targeted deposition. **j, l** 3D Model and printed part of the bellow actuator shown in (**i, k**) when subjected to a negative pressure, showing bending motion. Scale bars: (**a**) 6.5 mm, (**b**) 5.3 mm, (**c**) 6.4 mm, (**d**) 1 mm, (**e**) 6.5 mm, (**f**) 1 mm, (**g**) 6 mm, (**h**) 700 μm, (**k, l**) 5.1 mm.

average accuracy for the positive EmVP process of 0.23 ± 0.28 mm. We also show the embedding of fine features, such as filaments embedded in the wall of a hollow cylinder, Fig. 2e. A close up of the cross-sectional cut is shown in Fig. 2f. Finally, we show how embedding a stiff material in a soft bath can be used to generate a pneumatic actuator, Fig. 2i–l, which can deflect when subject to a negative pressure, shown in Supplementary Movie 2. An overview of the total print times and the duration of the separate EMB3D and TVAM sections of the process is provided in Supplementary Table 1. The characterization of the minimum printable positive features through our TVAM setup was carried out by printing increasingly smaller structures, shown in Supplementary Fig. 6, showing a minimum feature in the 300 μm range. For the EMB3D step, the diameter of the deposited liquid ink thread or filament was 175 μm, shown in Supplementary Fig. 7c, d.

### Negative EmVP
Direct TVAM printing of negative features allows to produce microchannels through a rapid process, but typically requires careful tuning of the projection patterns, especially with regards to diffusion effects. In our TVAM setup, the minimal channel diameter achievable is in the range of 500 μm, even for simple, straight 4 mm long designs, as shown Supplementary Fig. 4. To make the generation of smaller channels more accessible, we developed a complementary process to the positive EmVP process presented above, negative EmVP, shown in Fig. 1g, h and Fig. 3. Here, the support bath holds a liquid sacrificial inks in-place before photopolymerization by VAM, which cures a structure around the deposited sacrificial inks. As the print progresses, only the supporting bath is cured, and when the print is complete the unreacted sacrificial inks can be flushed out. The channel size is thus chiefly controlled by the diameter of the deposited liquid thread and needle size. Pluronic PE3100 was chosen as the sacrificial ink for its low viscosity and surface energy. Figure 3a shows the model of the printed 2.5D chip, consisting of a simple Y channel in a rectangular chip. As both EMB3D printing of a channel and VAM of the surrounding structure are very fast, we can produce open microchannels in structures in the cm-scale in less than 3 minutes. Thanks to the rheological properties of the supporting bath, the liquid ink is held in place, as shown by the images in Fig. 3b, f taken during the VAM print process. Figure 3c, g shows the final structures with inner channels, namely a flat Y-junction (Fig. 3c) and a cylindrical 3D chip (Fig. 3g) containing

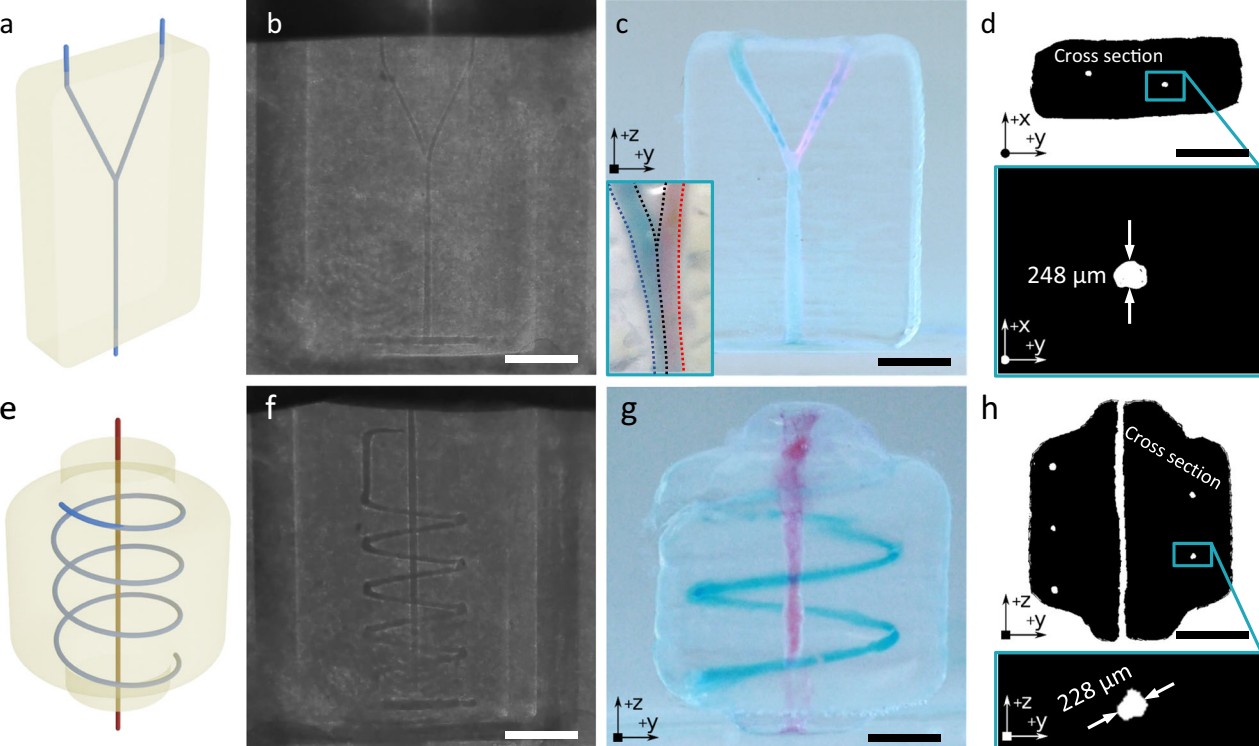

**Fig. 3 | Negative EmVP process for the printing of 3D microfluidic chips with small channels. a** 3D model of a flat Y-junction chip. **b** Shadowgram taken during VAM printing of the junction with the channel outline clearly visible. **c** Final Y-junction corresponding to the model shown in (**a**). The inset show co-flow when extruding red and blue died water through the two inlet arms of the chip. **d** Horizontal cross sections extracted from CT scans of the chip in (**c**), and inset showing channel diameter of 248 μm. **e** Model of a cylindrical chip with two channels embedded in it (**f**) shadowgram taken during the VAM printing of the chip with the outline of the channels deposited in sacrificial ink clearly visible. **g** Final chips filled with died water in blue and red. **h** Vertical cross-sectional image obtained from microCT scanning showing the channel diameter of 228 μm for the helical channel in the cylindrical chip in the inset. Scale bars: (**b**, **f**) 4.5 mm, (**c**) 3.7 mm, (**d**) 4 mm, (**g**) 2.2 mm, (**h**) 3 mm.

two separate channels where the external one is coiled around the central one. The inset in Fig. 3c shows typical co-flow behavior of two colored inks when extruded at the same time. To determine the channel size, we executed microCT scans of the printed parts and extracted the channel cross sections. The minimal channel diameter obtained through negative EmVP using a 150 μm diameter needle was 119 μm, shown in Supplementary Fig. 5. This is a definite improvement over pure TVAM in our setup, with which at 500 μm diameter only simple channels can be produced (see Supplementary Fig. 4).

## Rheological and optical characterization

To determine the curing behavior of the Mat 1 and Mat 2 materials used to produce the prints of Fig. 2 as well as the chips in Fig. 3, we performed photorheological tests, shown in Fig. 4a, b. Two initiator concentrations were selected that result in similar gel times so that both sides of the object reach gelation with similar energy doses. In our case, Mat 1 reaches gelation after 59.8 s from the start of the measurement, while Mat 2 reaches gelation after 64.7 s. The absorption coefficient at the working wavelength of 450 nm of the two resins is also characterized to use as input in the projection generation process, and confirming that both allow light to travel in the vial. The resulting absorption coefficient curves are shown in Fig. 4c. To assess the supporting properties necessary for EMB3D as well as printability of the Mat 2 bath and Mat 1 ink, we perform stress/recovery tests, Fig. 4d, e, by alternating high and low shear steps and recording the change in storage and loss modulus. The recovery time is then given by the time elapsed between the removal of the shear stress and the recovery of the bath to solid behavior. Mat 2 shows a low recovery time of 1.7 s between the removal of the stress and the recovery of the solid

behavior in the supporting bath. Completing the characterization of the chosen materials, the shear rate-viscosity relationship is shown in Fig. 4f, showing that the developed Mat 1 is shear thinning and possesses a recovery time of 4.7 s, shown in Fig. 4e which helps in obtaining a smooth extrusion and improving feature definition during the EMB3D process. In Supplementary Fig. 2 and Supplementary Fig. 3 we characterize the difference in rheological behavior for different loadings of the rheological modifier in Mat 2, as well as the amplitude sweeps for Mat and Mat 2, both showing clear yielding behavior.

## Discussion

Creating multi-material volumetric prints has been shown to pose significant challenges due to the part curing all at once, instead of layer-by-layer like in traditional 3D printing. Several techniques have been attempted, each with their advantages and disadvantages. The techniques can be categorized into vat-exchange processes, where the material in the print vial is exchanged after each volumetric printing step, and embedding processes, where additional materials are embedded into the print vial before printing and only a single volumetric printing step is performed. Vat-exchange processes have been shown to produce parts containing spatially varying materials, such as in sequential multi-material volumetric printing (SMVP)[28,29]. Here, for each material, a set of projection is used to cure a structure, and the excess material is removed to make space for the next material in the sequence. This requires additional support structures to maintain the positioning of the part during the material exchange step. The need for support structures that consecutively need to be removed leads to the loss of one of the significant advantages that TVAM has over traditional 3D printing – which is support free printing. SMVP also

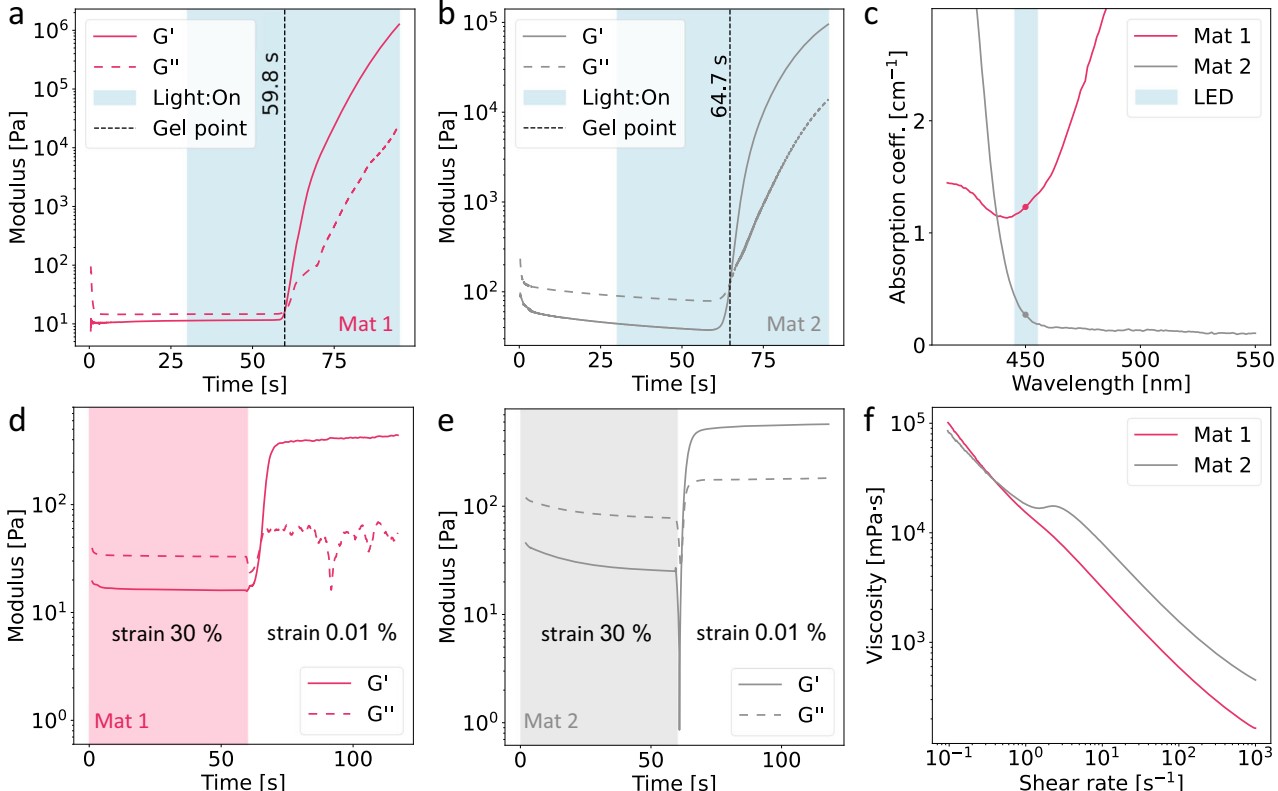

**Fig. 4 | Photorheological, rheological and optical characterization of the supporting baths and inks used in this work. a** Photorheological test of Mat 1, showing gelation being reached at 59.8 s from the start of the measurement. **b** Same test as in (**a**) for Mat 2, showing similar time to gelation of 64.7 s. As the time to gelation are similar, both materials cure at similar times when undergoing VAM together. **c** Absorption coefficient of the two materials, highlighting values at the VAM printer LED emission wavelength. The low absorbance of Mat 2 makes it suitable for VAM, while Mat 1 would normally have too high absorbance if used directly for VAM, but thanks to being only embedded in specific regions of the volume, it can still be used. **d** Stress/recovery test of Mat 1, showing a single high-shear cycle. The recovery time for Mat 1 is 4.7 s. **e** Same test as in (**d**) for Mat 2, showing a low recovery time of 1.7 s, making it suitable as a support bath for EMB3D. **f** Flow curves of Mat 1 and Mat 2, showing shear thinning behavior for both. For Mat 1, this aids in obtaining a smooth extrusion flow during EMB3D.

requires precise alignment between the part printed in each step and the set of projections used in the following step. This is challenging to achieve when adding materials to the vial containing previously printed structures, as the added material is likely to easily deform the printed structure. This is further complicated by the softness of the parts printed by TVAM before postcuring. Aligning the part and projection was performed manually, by adding a second wavelength to the volumetric printer orthogonal to the curing wavelength of the materials, increasing cost and complexity of the system. This secondary wavelength used for alignment also needs to have non-zero transmittance for all materials used in the SMVP process. Having multiple projection sets, one for each step, also increases the computational cost of a print.

Embedding processes, such as Embedded Extrusion-Volumetric Printing (EmVP) rely on embedded 3D printing (EMB3D) to precisely embed a photopolymerizable ink material into a photopolymerizable support bath[29,30,38]. This eliminates the need to print support structures, as there is no material exchange step. The support bath holds the embedded ink in place, making alignment significantly simpler. The only requirement for alignment is to maintain the orientation of the vial between the EMB3D step and the TVAM step. Despite the clear benefits EmVP can bring to volumetric multi-material printing, it has so far only been used to produce structures where the volumetrically printed part encases the embedded ink in an overprint process. This is because it requires polymerizable embeddable inks that cure at the same time as the polymerizable support bath to allow the simultaneous volumetric printing of both materials. The design of such materials with matched curing properties is challenging.

In the positive EmVP process presented here, we show that by engineering the gelation time of the materials to be similar (Fig. 4), we can produce multi-material parts (Fig. 3). We achieve this by first defining the placement of the different materials, and thus material properties, through EMB3D. We then concurrently polymerize the materials into a single, multi-material structure where either material can be arbitrarily positioned in the bulk or be exposed at the surface. This is an advancement compared to existing EmVP approaches that show overprinting, which is characterized by fully enclosing one material in the bulk of another material. Extending EmVP with the use of sacrificial inks we also show a negative EmVP process: by embedding a non-polymerizable ink into a polymerizable support bath via EMB3D, we can use TVAM to structure microfluidic chips that would have otherwise been impossible to print with the TVAM printer alone. TVAM has been used to accurately reproduce simple, straight microchannels with reported diameters as low as 124 μm with the use of high-resolution volumetric printers while for more complex channels the achieved channel diameters in the range of 250-500 μm[23,29,39,40]. A comparison of multi-material methods developed for TVAM showing advantages and disadvantages is summarized in Supplementary Table 2. In the case of the printer used in this work, diffusion effects and overcuring make printing negative features lower than 500 μm challenging, as shown in Supplementary Fig. 4. Using negative EmVP however, we were able to produce channel diameters down to 119 μm by sidestepping the limitations of a pure TVAM process through

non-polymerizable sacrificial inks. The negative EmVP process is also more scalable than a pure TVAM process for producing channels with small diameters in a larger scale print: in EmVP the channels diameter can be decreased by using custom needles with small diameters. To achieve small channel diameters in a pure TVAM process, the pixel size has to be reduced significantly to increase resolution. This requires not only additional optics but also reduces the build volume. When setting up an EmVP process, some points need to be considered: we found that it is beneficial to have the embedded ink represent the part of the print with the lower volume as well as having the higher absorption. This allows to minimize the duration of the EMB3D step, as well as improve the TVAM step given the limited extent of the higher-absorbing material. We also note that when having a mismatch in cure time, the faster curing material being the embedded one helps obtaining prints with acceptable surface error, as once the embedded ink is polymerized, overcuring of the surrounding support bath starts when the remaining areas of the structure polymerize.

In conclusion, the positive and negative EmVP processes presented in this paper show how EMB3D and TVAM can be used as complementary processes. They enabled the integration of regions with widely different material properties into a single structure. Through positive EmVP, we showed prints composed of regions with different tensile properties. Through negative EmVP, we extended the capability of TVAM to generate microscale channels beyond the capabilities of the volumetric printer when used in a plain TVAM process. We achieved this by using a sacrificial ink to directly embed the desired channel in the support bath. Negative EmVP reduces reliance on complex modeling of material behavior and projection optimization, while maintaining a simple, single lens TVAM print setup. In EmVP, costly modifications such as multi-wavelenght projection systems are not required to align the embedded ink to the projections. This is a significant advancement for the field of TVAM and will allow to increase the complexity of TVAM prints, expanding the possible applications of TVAM in microfluidics and multi-material printing.

## Methods

### Materials

Phenylbis (2,4,6-trimethylbenzoyl)-phosphine oxide (BAPO), Camphorquinone (CQ), Ethyl-4 dimethylamino benzoate (EDAB), Rhodamine B and 1,6-Hexanediol Diacrylate (HDDA) were purchased from Sigma Aldrich GmbH (Germany). Aerosil R805 was kindly provided by Evonik GmbH (Germany). Aliphatic Urethane Acrylate (Genomer 1122TF) was kindly provided by Rahn AG (Switzerland). Pluronics PE3100 was purchased from BASF GmbH (Germany). All chemicals were used as received.

### Resin and matrix preparation

For the soft support bath (Mat 2), 0.5 wt% of BAPO was pre-dissolved in acetone in a polypropylene vial and 8 wt% Aerosil R805 was added. Then, Genomer 1122TF was added. The resin was mixed with a Hausschild Speedmixer Benchtop (Hausschild GmbH, Germany) for 5 minutes at 3500 rpm to homogenously disperse the silica, and subsequently portioned into separate glass printing vials. For the stiff resin (Mat 1), the same general process was carried out using HDDA as the monomer and 0.25 wt% CQ and EDAB as the initiator. To adjust rheological properties, 7 wt% Aerosil R805 was used. Additionally, 0.015 wt% of Rhodamine B was added to Mat 1 to aid in visualizing material differences.

### Embedded Extrusion-Volumetric Printing

The Embedded Extrusion-Volumetric Printing (EmVP) process is carried on a pair of custom-built 3Dprinters. For EMB3D, a 3-axis cartesian-style system is used, based on standard 400 steps/rotation nema 17 motors for motion and a non-captive nema 11 motor used as the extruder, accepting standard 2 ml syringes with luer-lock needles,

controlled by a FLY Gemini V3 board running Klipper. For VAM, we use a VAM printer design based on a LCD with a pixel size of 50 μm and a working wavelength of 450 nm[15]. Both printers are controlled by a combination of Klipper and custom python scripts[41]. The projections for VAM are generated following a modified previously reported process and optimized by Object Space Model Optimization (OSMO) modified according to the Proportional-Integral Histogram Equalization technique[42,43]. The projection generation is described in more detail below. Briefly, the desired 3D model as STL is voxelized with a voxel size matching the VAM printer resolution, then forward projected to obtain an initial set of projections, which is the refined by calculating the expected deposited volumetric dose and optimized by comparing the reconstructed dose with the initial model. Once the projection generation is completed, the images are exported as a stack of PNG images which are then loaded on the VAM printer. The section(s) of 3D model to be printed in a different material are loaded in a slicer, in our case SuperSlicer, to obtain the gcode reproducing the partial structure[44]. The gcode is then sent to the EMB3D printer. A cylindrical 5 ml glass vial is filled with the polymerizable support bath and loaded on the EMB3D printer, where the embedding of the desired structure is performed. All EmVP prints are performed with a 0.4 mm internal diameter needle, setting the line width and height to 0.32 mm for the positive process and to 0.25 mm for the negative process, unless noted otherwise. Embedding of Mat 1 is performed with a print speed of 15 mm s⁻¹, while the embedding of Pluronic PE3100 for the generation of microchannels is performed at 5 mm s⁻¹. When the EMB3D print completes, the vial is moved to the VAM printer, taking care to maintain alignment between reference frames. The VAM print is then triggered, solidifying the desired structure based on the optimized projections. The VAM print is deemed completed when the shadowgrams, cast by a 620 nm LED light source placed orthogonally to the VAM light propagation direction, match with the desired shape. After printing is completed, the part is gently removed from the vial and placed in a glycerol bath to remove uncured resin and postcured in a XYZPrinting UV Curing Chamber (XYZPrinting, China) for 5 minutes. In the case of negative EmVP, the channels left by the sacrificial ink are flushed with isopropanol and compressed air before postcuring to avoid clogging due to leftover resin.

### Projection generation

The selected 3D model is split into regions to be printed in the different materials. A volume representing the vial is then created. Each part of the split model is then added to the vial volume, replacing the absorbance values to match those of the material they will be printed in. Finally, the 3D model is voxelized and forward projected while accounting for the absorption of the combined vial volume. Once an initial set of projections is obtained, the inverse process is carried out to obtain an estimate of the deposited energy dose. For each step of length $d_i$ along the light propagation direction $x$, the deposited dose $E_i$ is calculated as

$$E_i = d_i \cdot \alpha_i \cdot I_{i-1} e^{-\alpha_i d_i} \tag{1}$$

where $\alpha_i$ is the absorbance at the voxel point in consideration and $I_{i-1}$ is the transmitted intensity for the previous step, or the initial intensity in the first step. Usual volumetric printing optimization algorithms such as PIHE to refine the projections for printing are used. The projection generation, implemented in Python, is carried out on the bwUniCluster shared cluster running an Intel Xeon Gold 6230 and Nvidia Tesla V100. Forward, backward and optimization example code is available as Supplementary Code 1.

### Rheology and photorheology

The photorheological properties of the resins where characterized using a MCR301 rheometer (Anton Paar GmbH, Germany) fitted with

a glass bottom plate to allow light exposure during the rheological measurement. All measurements were performed with plate-plate geometry using a gap distance of 0.1 mm, unless specified. The rheological behavior of the support bath is characterized with regards to storage (G') and loss (G") moduli by observing their change as an increasing oscillation amplitude is applied, from 0.01% to 100%, as well as by observing the change in viscosity against shear rate from $0.1\,s^{-1}$ to $1000\,s^{-1}$. The thixotropic time was measured on the same machine by a three-step interval test, where the sample is subjected to a constant amplitude oscillation of 30% for 60 s followed by a 0.01% amplitude period of 60 s. This is repeated 2 times for a total measurement duration of 240 s. The thixotropic time is given by the delay between the reduction in oscillation amplitude and the switch of behavior of the support bath from liquid to solid. Finally, the gel point was measured by applying a constant amplitude oscillation of 30% with a frequency of 5 Hz. The amplitude was chosen to ensure that the support matrices are in liquid behavior (G" > G') at the start of the photorheological measurement. After 30 s from the start of the measurement to allow for the initial values to stabilize, an LED is switched on and the resin is exposed to 450 nm light with an intensity at the rheometer plate of $5.8\,mW\,cm^{-2}$. The illumination continues during the rest of the measurement and the light exposure causes polymerization of the resin being tested, inducing a transition from liquid to solid behavior after a gel time, defined by the crossing point of G' and G".

### UV Vis analysis
The light absorbance of the formulations was measured with a UV Vis spectrophotometer Thermo Scientific Evolution 201 UV-Vis-spectrophotometer (Germany) from 400 to 550 nm, using PMMA cuvettes with a side length of 1 cm.

### microCT scanning
The microCT scanning was performed on a miniCT machine (PXR GmbH, Germany) with an axial scanning path and a cone-beam emitter with a 20 µm spot size. The scans were performed at 0.62 s exposure with a 10 W power setting and a voxel size of 15 µm. After the scans where completed, the resulting projections were processed to reconstruct the shape of the scanned parts with the X-AID software, obtaining a voxel volume with a voxel size of 15 µm and further processed using Slicer3D to extract cross sectional images.

### Hausdorff distance
The Hausdorff distance was obtained by using CloudCompare Cloud-To-Mesh Distance function between the 3D model of the structure and the surface of the printed structure, extracted from a CT scan with 3D Slicer[45–47].

### Shore D hardness testing
The hardness of printed samples of Mat 1 and Mat 2 where determined using Shore D hardness measurement device (Zwick GmbH, Germany). Each sample was probed on four different spots. Each measurement lasted 15 s.

### Tensile testing
The tensile properties of cast samples of Mat 1 and Mat 2 where determined using an universal testing machine Inspekt Table (Hegewald & Peschke GmbH, Germany) following a scaled down ISO 527-1 norm.

## Data availability
The data generated in this study are provided in the Source Data file. Additional data are available from the corresponding author upon request. Source data are provided with this paper.

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

## Acknowledgements

The work reported here was funded by the German Federal Ministry of Research and Education (BMBF) within the NanoMatFutur Program project 03XP0299 (D.H.). The authors acknowledge support by the state of Baden-Württemberg through bwHPC and by the Freiburg Materials Research Center (FMF).

## Author contributions

S.T. and D.H. conceived the idea. S.T. designed the experiments. S.T., G.V., Q.S. and N.N. performed the experiments. S.T. analyzed the results. S.T. and D.H. drafted the manuscript and all authors contributed to writing the manuscript.

## Funding

## Competing interests

The authors declare no competing interests.
