## [Transparent Peer Review file · Nature Communications]

Additive manufacturing of multi-material and hollow structures by Embedded Extrusion-Volumetric Printing

Corresponding Author: Dr Dorothea Helmer

Version 0:

Reviewer comments:

Reviewer #1

(Remarks to the Author)

The authors describe the preparation of multi-material 3D structures using a combination of embedded extrusion printing and tomographic volumetric additive manufacturing. The technology has been already described in previous works and so it is its application in producing complex multi-material structures, but the types of materials used in this specific manuscript are new for the specific application of Embedded extrusion Volumetric Printing. Considering the relatively limited amount of materials currently available for tomographic printing, the data is of interest for the materials science community.

-The same technology was described before in 2023 (doi: 10.1002/adma.202301673), and later applied also in other studies, also with non-granular baths (doi: 10.1002/adma.202309026; 10.1101/2024.09.21.614231). This is not very clear in the introduction, as the authors do cite the first paper, but afterwards state that “the generation of multi-material freestanding objects that comprise of material areas with distinct material properties in a single part rather than overprinted inner structures have not yet been reported”, while this has indeed been reported in the three references listed above. The introduction and discussion sections should be adjusted accordingly. The authors should also clarify better in the text where the novelty of their work stands. The discussion in particular, is especially weak. As of now, it is just a summary of the results section. There is no real discussion on the implications or contextualization with the rest of the body of the literature, and it gives the interpretation that this is the first time this approach is applied, which is not the case.

-Moreover, it is questionable whether is necessary to invent yet another acronym (VAM-EMB3D, ETVAM), since the technology has already been described, known and introduced as Embedded Volumetric Printing (EmVP), and the approach presented in this new manuscript is essentially a new application of the same technology. Better to avoid the generation or redundant names.

-the authors indicate thixotropy as an ideal property for support baths. This is not precise, however. Thixotropy indicates a time-dependent change of viscosity at a constant shear stress. what is rather important for support baths is to have a clear yield stress behavior, and an elastic recovery rate that matches the deposition speed set by the extrusion nozzle's translation velocity.

- the rationale for the subtractive manufacturing section is unclear. The structures produced with the fugitive inks could be easily produced with tomographic printing alone, in fact, one strength of this technology is the ability to produce complex channels, that cannot be easily produced by extrusion (of sacrificial inks). The use of embedded extrusion seems rather a redundancy or limitation compared to the capacity of tomographic VAM, and the authors should explain better the limitations and advantages they see for the technology they propose.

Reviewer #2

(Remarks to the Author)

This manuscript from Tisato and colleagues about embedded tomographic additive manufacturing (ETVAM) describes a combined approach to 3D printing multimaterial structures by using tomographic VAM and embedded extrusion printing into a support bath. This is a useful addition to the growing range of capabilities for the tomographic volumetric AM paradigm. The authors demonstrate two modes, which they term additive ETVAM (in which the extrusion needle deposits a second material that cures to become part of the final structure), and subtractive ETVAM (in which the extrusion material is sacrificial and flushed away after final printing).

Overall, the work is well-executed, and the results are mostly clearly presented and described. However, for a journal with the broad audience of Nature Communications, I would hope for some more generality to be presented to readers, or some broader design principles beyond this initial demonstration. So I would recommend acceptance and publication of this work, but only after substantial revisions to answer at least some of the following questions:

1. How does the resolution and accuracy of this approach, which combines EMB3D with TVAM, compare to the resolution and accuracy of each technique on its own? Are there limitations on geometric shapes/flexibility (one would imagine that many limitations of extrusion-based embedded printing remain, but only for one of the materials)? While a truly comprehensive study is probably too large in scope for an initial paper, the authors likely understand the overall trends to provide some comments and discussion for a reader interested in using this technique, but perhaps not with identical materials and formulations or printing conditions. Are there any principles that can be described, such as sources of distortion, feature blurring/minimum spacing? In Figure 3, for instance, the resulting channels are clearly distorted from a cylindrical/circular cross-section. How does such distortion compare to other embedded extrusion technologies?

2. Relatedly, the authors mention that after EMB3D printing, "the vial is moved to the VAM printer taking care to maintain alignment between reference frames." How is this alignment done? If by hand, what are the limitations on accuracy of alignment? If not by hand, please describe this.

3. For additive ETVAM the authors talk about "similar gel times" for the two materials. How closely must these be matched? Here we see a difference of ~8% in terms of exposure time, but what are the consequences of a greater mismatch and what is the threshold? Is it important to match gel *times* or absorbed volumetric optical dose? How do the absorbances of the resins compare for the 0.5 wt% BAPO and 0.25 wt% CQ?

There are also a few minor corrections that I would suggest:

- I'm not a fan of the "additive" vs. "subtractive" terminology for ETVAM, because additive is already used as an adjective on "additive manufacturing" in contrast to subtractive manufacturing, which is milling, cutting, drilling, etc. Perhaps "positive ETVAM" and "negative ETVAM" might be less over-loaded terms?

- How was the silica loading % chosen? Was any sort of parameter sweep carried out to determine an optimal value? What is, for example, the minimum loading required to have these materials behave as desired for ETVAM?

- It would be excellent if the authors could at least estimate the Young's modulus values for Mat 1 and Mat 2, in addition to the Shore D durometer values, as the latter are difficult to compare to mechanical properties of other materials reported in the literature (durometer is significantly less frequently reported).

- What is "SH resin" mentioned on Line 271? What is PLU mentioned on Line 293? Also, Pluronic does not have an "s" at the end of its name.

- On line 281, the authors make reference to "product-derivative histogram equalization" from Ref 39, but that's incorrect (it's Proportional-Integral). Please check how Webber et al describe PIHE. Similarly, when mentioning PIHE on line 313, please cite ref 39.

- When observing whether a shadowgram for the VAM print, as described in lines 297-298 "matches with the desired shape," how is this measured? Using a camera? By comparison to an image/cross-section of the input model?

Reviewer #3

(Remarks to the Author)

The authors show the combination of embedded extrusion and tomographic VAM to jointly cure two materials to produce 3D prints with selective material properties in the volume.

The authors show many examples, which are well realized and interesting.

The idea of using embedded 3D printing and VAM has been demonstrated previously in the context of bioprinting by Ribezzi et al "Shaping Synthetic Multicellular and Complex Multimaterial Tissues via Embedded Extrusion-Volumetric Printing of Microgels" Adv. Mat. 2023. The study describes the use of embedded printing and VAM to produce multi-material constructs. This study is referenced by the authors but it should be cited separately and the authors should explicitly highlight the difference with their work. In my opinion, the difference that the authors bring is that the embedded material is also polymerizable with the same gelation time as the matrix material and that they are both shaped by VAM resulting in a 3D print with the two material having very different mechanical properties.

From the examples given in additive ETVAM, it seems that all of them could have been realized with two material overprinting. The latter likely with more flexibility. Can the author describe what is the real advantage of their method?

The light absorbance of the two materials is widely different, 30% vs 80% in figure 4 c. Is it measured for the same material length. It would be more indicative of giving the absorption of the material + Photoinitiator in units of 1/cm.

It is not clear if the pattern computation takes into account the light absorption of the combined material. In other words, it seems that the 3D location of the first material needs to be known in the print in order to account for the propagation of light which can traverse both material.

Is the index of refraction difference between the two material taken into account in the computation of the pattern ? A small difference would make the rays deviate from a straightline. This would compromise resolution.

The gelation time is measured from rheology, however it is known in VAM that diffusion tend to increase the gelation time for smaller structures. Given the two different material light absorption, it would seem difficult to make arbitrary 3D structures for both materials simultaneously. The authors did not comment on resolution and the ability to generate arbitrary structures.

Version 1:

Reviewer comments:

Reviewer #1

(Remarks to the Author)

The authors carefully revised the manuscript and addressed the previous concerns.

Reviewer #2

(Remarks to the Author)

I very much appreciate the authors' thorough response. In my opinion they have adequately addressed all the comments and suggestions made by the referees. I particularly appreciate the addition of Figure S8 to quantitatively assess the printing accuracy of the final structure in comparison to the source model. I recommend publication of the revised manuscript.

Reviewer #3

(Remarks to the Author)

From the reviewers questions, all have indicated that explaining better the difference between the proposed embedded extrusion and VAM with respect to other works demonstrating embedded extrusion is essential.

To me, this is important for the readers to understand which method to use when combining embedded extrusion with VAM. There are advantages and disadvantages. I believe a table highlighting the pros and cons would be very useful and clear a possible confusion

Response to the reviewers – Tisato et al., “Additive manufacturing of multi-material and hollow structures by Embedded Extrusion-Volumetric Printing”

Dear reviewers, dear editors,

we thank the reviewers for the detailed review, the expert input on our work and the helpful suggestions they have made. We have addressed all the suggestions and issues raised – please find our response to the comments of the reviewers on the following pages. The reviewer’s comments are set in *italic* whereas our comments are set in upright font. Where relevant, manuscript excerpts in Times New Roman with changes **highlighted in yellow** have also been added here for clarity.

Reviewer #1:

The authors describe the preparation of multi-material 3D structures using a combination of embedded extrusion printing and tomographic volumetric additive manufacturing. The technology has been already described in previous works and so it is its application in producing complex multi-material structures, but the types of materials used in this specific manuscript are new for the specific application of Embedded extrusion Volumetric Printing. Considering the relatively limited amount of materials currently available for tomographic printing, the data is of interest for the materials science community.

Answer: We thank the reviewer for the assessment of our work and the thorough review.

Question 1.1: *The same technology was described before in 2023 (doi: [10.1002/adma.202301673](https://doi.org/10.1002/adma.202301673)), and later applied also in other studies, also with non-granular baths (doi: [10.1002/adma.202309026](https://doi.org/10.1002/adma.202309026); [10.1101/2024.09.21.614231](https://doi.org/10.1101/2024.09.21.614231)). This is not very clear in the introduction, as the authors do cite the first paper, but afterwards state that “the generation of multi-material freestanding objects that comprise of material areas with distinct material properties in a single part rather than overprinted inner structures have not yet been reported”, while this has indeed been reported in the three references listed above. The introduction and discussion sections should be adjusted accordingly.*

Answer 1.1: While the publications mentioned by the reviewer are very impressive and show a combination of embedded printing and VAM, they show overprinting – the integration of one material that is fully enclosed in another material, similar to the integration of solid occlusions shown in early work from Kelly *et al* ([10.1126/science.aau7114](https://doi.org/10.1126/science.aau7114) , Figure 4). They also show sequential multi-material printing ([10.1101/2024.09.21.614231](https://doi.org/10.1101/2024.09.21.614231) , Figure 4) of two materials by printing one, then washing out uncured resin, adding a second material to the vial and printing it. However, these sequentially printed structures required additional support structure to help maintain alignment between fill steps.

The differences between our work and the mentioned publications are explained here in further detail:

- in [10.1002/adma.202301673](https://doi.org/10.1002/adma.202301673) a granular hydrogel made from GelMA “ μ Resin” is used as a support bath for cell-loaded bioinks (methyl cellulose, gellan gum/PEGDA inks). The μ Resin supports the printed bioink and around the deposited bioink the μ Resin is solidified using VAM. All deposited materials (bioinks) are fully embedded in the support material in the final part, qualifying it as overprinting.
- in [10.1002/adma.202309026](https://doi.org/10.1002/adma.202309026) gelatin is used to rheologically modify the printing materials to allow for printing of low-viscous inks. Inks are mixed with gelatin and VAM is used to solidify the ink/gelatin mixtures. Gelatin may be removed afterwards. This is also shown for integration of a second ink/gelatin mixture into the first ink/gelatin bath by embedded printing. Consecutively, the ink/gelatin bath is cured around this structure using VAM. This way, the integration of thin lines of approximately 300 μ m to 500 μ m diameter is shown. However, as above, the embedded material is fully surrounded by support material after the volumetric printing step and hence this is also an example of overprinting.
- [10.1101/2024.09.21.614231](https://doi.org/10.1101/2024.09.21.614231) shows bioprinting with a library of different molecular weight GelMA variants, using low molecular weight GelMA-based support baths. The final structures obtained via EmVP are all overprints, with the embedded ink fully immersed in the VAM print. The publication also shows a sequential technique for multi-material printing (sequential multi-material volumetric printing SMVP) where one material is printed by VAM, the unreacted resin is removed, new resin is poured on top and a different structure is solidified via VAM, in a sequential overprinting process. While this is quite interesting, it is a vat-exchange process and it presents significant challenges with keeping a consistent alignment of the partially printed structures between washing and filling steps, especially considering the soft nature of VAM prints without postcuring.

We therefore agree that overprinting has been widely shown in the literature. It holds many challenges, such as alignment of the structures and differences in refractive indices of the polymerized and unpolymerized regions of the print. In our work we show the simultaneous volumetric printing of different materials in one printed structure, where each material is free-standing and not embedded into the other material. This approach needs to be distinguished from the overprinting methods and has another set of challenges. For overprinting, the materials gelation times do not need to be matched: considering the embedded areas as an occlusion and overprinting around them, followed by curing at a later time is sufficient. In our approach, we needed to engineer a precise material match with regards to gelation time. The matching materials in terms of gelation time is demanding, because it needs to consider differences in the reactivity of the resin, for example when materials with a different number of functional groups are used. In the presented work, we matched the materials and show the printing of multi-material structures without vat exchange or overprinting.

We do, however, agree with the reviewer that we should make clearer what the differences of our approach are compared to the state of the art. Therefore, we have

- changed the respective sentence “*the generation of multi-material freestanding objects that comprise of material areas with distinct material properties in a single part rather than overprinted inner structures have not yet been reported*” to make this clearer to the reader.
- changed the introduction to include all mentioned citations and explain the state of the art in overprinting.
- To further prove that we do not show overprinting in this work, we have added Figure S1 to the supporting information, showing that after curing in positive EmVP, the embedded, colored material remains in the vial after extraction of the print, as the volumetric printing step cures a structure inside the embedded material

We have made the following changes to the manuscript, extending the introduction to include an improved explanation of the state-of-the-art:

However, multi-material 3D printing is still a challenge in VAM. There have been efforts to produce parts with spatially varying properties through dual-wavelength systems or stacking layers of different resins, as well as embedding preexisting parts in the resin and printing around them.²⁶⁻²⁹ However, the generation of multi-material freestanding objects that comprise of material areas with distinct material properties in a single part rather than overprinted inner structures have not yet been reported. Current methods for generating multi-material prints in TVAM include integrating a previously fabricated structure into a TVAM vial and curing around the structure,²⁷ generating stiffness gradient materials by selectively curing an acrylate-epoxy mixture in a dual-wavelength setup,²⁶ and varying material in the vertical direction by stacking different material layers.²⁹ Another method is sequential multi-material volumetric printing (SMVP), where an initial structure is printed in a first material, then the excess resin is washed out and the vial is filled with a second material which is then used to perform volumetric printing of the second part of the structure, similar to a vat-exchange process.^{29,42} These methods significantly broadened the capabilities of volumetric printing, but require complex alignment steps after resin exchange and additional support structures or are limited to mixtures of materials with orthogonal curing wavelengths requiring multi-wavelength volumetric printers. In terms of true multi-material prints the combination of EMB3D is a promising technique to achieve the precise deposition of materials in 3D and consecutively cure multi-material prints with TVAM. A promising method for multi-material VAM is the combination of embedded 3D printing (EMB3D) and TVAM, which is termed embedded extrusion-volumetric printing (EmVP).²⁸ Embedded 3D Printing (EMB3D) EMB3D is a material extrusion technique which has been developed to reduce the need for support structures common in other material extrusion techniques.³⁰ that deposits an ink into a supporting bath using a needle.³⁰ This is achieved by depositing an ink into a supporting bath that possesses specific rheological properties such as thixotropy, which allows the needle depositing the material being embedded to traverse the support bath while at the same time keeping the deposited liquid ink in place until the completion of the print. The supporting bath must possess specific rheological properties that allow the needle to traverse the support bath while at the same time keeping the deposited liquid ink in place until the completion of the print.³⁶ Finally, the printed parts can be polymerized all at once, when using polymerizable inks in

the additive EMB3D process.³¹ Alternatively, in the subtractive EMB3D process, the deposited ink is a sacrificial ink, while the support bath ~~is polymerizable, can be polymerized: and~~ once the bath is cured, the ink can be removed, which has been shown to produce microfluidics channels.^{32,33} Additive EMB3D, being a material extrusion process, can be easily used to produce multi-material structures, but requires long print times when producing complex structures, and can suffer from print accuracy issues due to the interaction between ink and support bath.³⁴⁻³⁶ EmVP has previously been used by Ribezzi *et al.* and Riffe *et al.* to bioprint multi-material VAM parts by embedding secondary material areas in the resin and overprinting bulk structures around them^{28,41,42} Taken together, all previously reported EmVP methods show overprinting, thus including one material - the ink of the embedded print or another object - in another material that is then polymerized by TVAM. None of the previously reported methods were able to produce multi-material structures in a single volumetric print step where different features of the structures are composed of different materials along the horizontal and vertical direction, without the need for support structures or vat exchange.

In this work, we show ~~an EmVP a combined VAM-EMB3D process~~ capable of precisely depositing an ink into a photopolymerizable supporting bath through EMB3D printing, followed by rapid VAM curing of the combined structure. When the embedded ink is ~~photopolymerizable and designed to cure at similar times as the support bath~~, this results in fast printing of multi-material parts ~~all with exposed, free standing surfaces by simultaneous volumetric curing of both materials thanks to the matched curing times~~, while when the embedded ink is not photopolymerizable we obtain 3D parts with precise embedded freeform microchannels. We present the development of the photopolymerizable supports baths and inks as well as the characterization of the printed multi-material parts and microchannels, showing several multi-material structures as well as microchannels with diameters down to less than ~~120230~~ μm .

41. Riffe, M. B. *et al.* Multi-Material Volumetric Additive Manufacturing of Hydrogels using Gelatin as a Sacrificial Network and 3D Suspension Bath. *Advanced Materials* **36**, 2309026 (2024).

42. Ribezzi, D. *et al.* Multi-material Volumetric Bioprinting and Plug-and-play Suspension Bath Biofabrication via Bioresin Molecular Weight Tuning and via Multiwavelength Alignment Optics. *Advanced Materials* **37**, 2409355 (2025).

The following figure was added to the supporting information as Figure S1:

Figure S1. Print process for positive EmVP and Comparison of prints in Mat 1, Mat 2 and combination of the two shown on prints of a spherical cage structure and showing proof that this approach is distinguished from EmVP overprinting. (a) Deposited Mat 1 in Mat 2 before curing. Note how the area of the deposited disk is fully filled. (b) Shadowgram of the positive EmVP of the spherical cage structure during the TVAM step. (c) Cured structure spanning across material boundaries. Note how the diameter of the printed structure is lower than that of the deposited disk of Mat 1. Thus, distinguishing this method from previously shown overprinting approaches. (d) After the structure is removed there is leftover Mat 1 in the vial, further proving that only the desired areas of Mat 1 are cured and that Mat 1 is not fully immersed in Mat 2 after the print and hence this approach is not characterized as overprinting. (e) Printed multi-material structure extracted from the vial. (f) Visual comparison of structures printed by TVAM in a single material vial of Mat 2 (left), Mat 1 (center) and a combination of the two (right). Scale bars: a,d 12 mm, b 5.5 mm, c 10.5 mm, e 6 mm, f 7.5 mm

Question 1.2: *The authors should also clarify better in the text where the novelty of their work stands. The discussion in particular, is especially weak. As of now, it is just a summary of the results section. There is no real discussion on the implications or contextualization with the rest of the body of the literature, and it gives the interpretation that this is the first time this approach is applied, which is not the case.*

Answer 1.2: We have rewritten and extended the discussion section to include more comparison to existing literature and how our work represents a significant advancement in the field of multi-material volumetric printing.

We have re-written the discussion to address the points raised by the reviewer:

~~We present here the ETVAM process, which combines the high speed of tomographic additive manufacturing with the support matrices of embedded 3D printing thus expanding the capabilities of VAM to produce freestanding multi-material 3D structures with areas of distinct material properties. We demonstrate an additive ETVAM process where additional photopolymerizable materials are selectively deposited in a photopolymerizable supporting bath and jointly cured to produce multi-material prints. The versatility of the process is shown by producing pneumatic actuators in a single print. We also demonstrate a subtractive ETVAM process to create freeform microchannels. We show several multi-material parts, using multiple embedding strategies proving the ink material can be deposited as desired, as well as examples of 3D structures with embedded microchannels, with channel diameters down to 228 μm . The additive ETVAM process enables the production of parts comprising materials with very different properties, such as different hardnesses, while by separating the chip geometry and channel geometry definition steps the subtractive ETVAM process enables the direct printing of freeform microchannels into a 3D structure in minutes. This process greatly extends possible application for VAM and enables novel ways of combining multiple materials into a single structure, as well as providing an extremely simplified process for the generation of microchannels in VAM printed structures.~~

Creating multi-material volumetric prints has been shown to pose significant challenges due to the part curing all at once, instead of layer-by-layer like in traditional 3D printing. Several techniques have been attempted, each with their advantages and disadvantages. The techniques can be categorized into vat-exchange processes, where the material in the print vial is exchanged after each volumetric printing step, and embedding processes, where additional materials are embedded into the print vial before printing and only a single volumetric printing step is performed. Vat-exchange processes have been shown to produce parts containing spatially varying materials, such as in sequential multi-material volumetric printing (SMVP).^{29,42} Here, for each material, a set of projection is used to cure a structure, and the excess material is removed to make space for the next material in the sequence. This requires additional support structures to maintain the positioning of the part during the material exchange step. The need

for support structures that consecutively need to be removed leads to the loss of one of the significant advantages that TVAM has over traditional 3D printing – which is support free printing. SMVP also requires precise alignment between the part printed in each step and the set of projections used in the following step. This is challenging to achieve when adding materials to the vial containing previously printed structures, as the added material is likely to easily deform the printed structure. This is further complicated by the softness of the parts printed by TVAM before postcuring. Aligning the part and projection was performed manually, by adding a second wavelength to the volumetric printer orthogonal to the curing wavelength of the materials, increasing cost and complexity of the system. This secondary wavelength used for alignment also needs to have non-zero transmittance for all materials used in the SMVP process. Having multiple projection sets, one for each step, also increases the computational cost of a print.

Embedding processes, such as embedded extrusion-volumetric printing (EmVP) rely on embedded 3D printing (EMB3D) to precisely embed a photopolymerizable “ink” material into a photopolymerizable support bath.^{28,41,42} This eliminates the need to print support structures, as there is no material exchange step. The support bath holds the embedded ink in place, making alignment significantly simpler. The only requirement for alignment is to maintain the orientation of the vial between the EMB3D step and the TVAM step. Despite the clear benefits EmVP can bring to volumetric multi-material printing, it has so far only been used to produce structures where the volumetrically printed part encases the embedded ink in an overprint process. This is because it requires polymerizable embeddable inks that cure at the same time as the polymerizable support bath to allow the simultaneous volumetric printing of both materials. The design of such materials with matched curing properties is challenging.

In the positive EmVP process presented here, we show that by engineering the gelation time of the materials to be similar (Figure 4), we can produce multi-material parts (Figure 3). We achieve this by first defining the placement of the different materials, and thus material properties, through EMB3D. We then concurrently polymerize the materials into a single, multi-material structure where either material can be arbitrarily positioned in the bulk or be exposed at the surface. This is an advancement compared to existing EmVP approaches that show overprinting, which is characterized by fully enclosing one material in the bulk of another material. Extending EmVP with the use of sacrificial inks we also show a negative EmVP process: by embedding a non-polymerizable ink into a polymerizable support bath via EMB3D, we can use TVAM to structure microfluidic chips that would have otherwise been impossible to print with the TVAM printer alone. TVAM has been used to accurately reproduce simple, straight microchannels with reported diameters as low as 124 μm with the use of high-resolution volumetric printers while for more complex channels the achieved channel diameters in the range of 250-500 μm .^{23,42-44} In the case of the printer used in this work, diffusion effects and overcuring make printing negative features lower than 500 μm challenging, as shown in Figure S4. Using negative EmVP however, we were able to produce channel diameters down to 119 μm by sidestepping the limitations of a pure TVAM process through non-polymerizable sacrificial inks. The negative EmVP process is also more scalable than a pure TVAM process for producing channels with small diameters in a larger scale print: in EmVP the channels diameter can be decreased by using custom needles with small diameters. To achieve small channel diameters in a pure TVAM process, the pixel size has to be reduced significantly to increase resolution. This requires not only additional optics but also reduces the build volume. When setting up an EmVP process, some points need to be considered: we found that it is beneficial to have the embedded ink represent the part of the print with the lower volume as well as having the higher absorption. This allows to minimize the duration of the EMB3D step, as well as improve the TVAM step given the limited extent of the higher-absorbing material. We also note that when having a mismatch in cure time, the faster curing material being the embedded one helps obtaining prints with acceptable surface error, as once the embedded ink is polymerized, overcuring of the surrounding support bath starts when the remaining areas of the structure polymerize.

In conclusion, the positive and negative EmVP processes presented in this paper show how EMB3D and TVAM can be used as complementary processes. They enabled the integration of regions with widely different material properties into a single structure. Through positive EmVP, we showed prints composed of regions with different tensile properties. Through negative EmVP, we extended the capability of TVAM to generate microscale channels beyond the capabilities of the volumetric printer when used in a plain TVAM process. We achieved this by using a sacrificial ink to directly embed the

desired channel in the support bath. Negative EmVP reduces reliance on complex modeling of material behavior and projection optimization, while maintaining a simple, single lens TVAM print setup. In EmVP, costly modifications such as multi-wavelength projection systems are not required to align the embedded ink to the projections. This is a significant advancement for the field of TVAM and will allow to increase the complexity of TVAM prints, expanding the possible applications of TVAM in microfluidics and multi-material printing.

43. Falandt, M. *et al.* Spatial-Selective Volumetric 4D Printing and Single-Photon Grafting of Biomolecules within Centimeter-Scale Hydrogels via Tomographic Manufacturing. *Advanced Materials Technologies* **n/a**, 2300026 (2023).

44. Viola, M. *et al.* Thermal Shrinking of Biopolymeric Hydrogels for High Resolution 3D Printing of Kidney Tubules. *Advanced Functional Materials* **34**, (2024).

Question 1.3: *Moreover, it is questionable whether is necessary to invent yet another acronym (VAM-EMB3D, ETVAM), since the technology has already been described, known and introduced as Embedded Volumetric Printing (EmVP), and the approach presented in this new manuscript is essentially a new application of the same technology. Better to avoid the generation or redundant names.*

Answer 1.3: We agree with the reviewer. In the interest of clarity, we have updated the name throughout the manuscript to "positive embedded volumetric printing" (positive EmVP), replacing "additive ETVAM", and "negative embedded volumetric printing" (negative EmVP), replacing "subtractive ETVAM". Positive EmVP refers to multi-material printed structures, with the embedded ink being photopolymerizable, while negative EmVP refers to creating channels in a VAM-printed structure, with the embedded ink being a sacrificial ink.

Question 1.4: *The authors indicate thixotropy as an ideal property for support baths. This is not precise, however. Thixotropy indicates a time-dependent change of viscosity at a constant shear stress. what is rather important for support baths is to have a clear yield stress behavior, and an elastic recovery rate that matches the deposition speed set by the extrusion nozzle's translation velocity.*

Answer 1.4: The reviewer is right, the yield stress and recovery rate are of high importance for supporting baths used in EMB3D and EmVP. As we performed a 3-Interval-Time-Test (3ITT) in oscillatory mode, the resulting curves describe the structure breakup/recovery time. We have updated our wording to more precisely describe the rheological properties, and added amplitude sweep and flow curves used to evaluate yield stress to the supporting information as Figure S3 and S2 respectively.

We have made the following changes to the manuscript:

To assess the supporting properties necessary for EMB3D as well as printability of the Mat 2 bath and Mat 1 ink, we perform stress/recovery tests ~~thixotropic time tests~~, Figure 4d-e, by alternating high and low shear steps and recording the change in storage and loss modulus. The ~~recovery thixotropic~~ time is then given by the time elapsed between the removal of the shear stress and the recovery of the bath to solid behavior. Mat 2 shows a low ~~recovery thixotropic~~ time of 1.7 s between the removal of the stress and the recovery of the solid behavior in the supporting bath. Completing the characterization of the chosen materials, the shear rate-viscosity relationship is shown in Figure 4f, showing that the developed Mat 1 is shear thinning and possesses a ~~recovery thixotropic~~ time of 4.7 s, shown in Figure 4e which helps in obtaining a smooth extrusion and improving feature definition during the EMB3D process. In Figure S2 and S3 we characterize the difference in rheological behavior for different loadings of the

rheological modifier in Mat 2, as well as the amplitude sweeps for Mat and Mat 2, both showing clear yielding behavior.

We have added the following figures to the supporting information:

Figure S2 Rheological characterization of the support bath for different R805 loadings. (a) Flow curves showing increased viscosity with increasing loading, all showing clear shear thinning behavior (b) Flow curves showing increasing shear stress for increasing R805 loading.

Figure S3 (a) Amplitude sweep test for Mat 2. (b) Amplitude sweep test for Mat 1. Both show clear yielding behavior.

Question 1.5: The rationale for the subtractive manufacturing section is unclear. The structures produced with the fugitive inks could be easily produced with tomographic printing alone, in fact, one strength of this technology is the ability to produce complex channels, that cannot be easily produced by extrusion (of sacrificial inks). The use of embedded extrusion seems rather a redundancy or limitation compared to the capacity of tomographic VAM, and the authors should explain better the limitations and advantages they see for the technology they propose.

Answer 1.5: We are happy to elaborate on the subtractive method (negative EmVP) further, to make this clearer. We disagree with the reviewer’s statement that small channels can generally be “*easily produced with tomographic printing alone*”. Volumetric printers generally struggle with diffusion effects and overcuring – and while overcuring can be exploited to produce thin channels (stopping the print right before the printed hole “closes”) this cannot be well controlled and a repeatable printing process is not possible. Using a sacrificial material that does not polymerize to create channels is therefore beneficial as it sidesteps these common TVAM issues with a simple process. The smallest channels created by VAM shown in the literature have a diameter of 124 μm for simple straight channels, and in the range of 250-500 μm for more complex channels ([10.1002/adfm.202406098](https://doi.org/10.1002/adfm.202406098)). We have integrated new data to show that our printer struggles to reproduce channels with diameters of 500 μm in a pure TVAM setup: an attempt of print thin holes is shown in Figure S4. Using negative EmVP however, our printer can easily produce $\sim 250 \mu\text{m}$ channels. To further emphasize the versatility of this approach, we reduced the needle size of the EmVP setup. This way we were able to produce channels down to $\sim 119 \mu\text{m}$, see Figure S5. This shows a) that using our EmVP process, our simple LCD-based printer setup can easily compete with more sophisticated machines. It further shows that b) our negative EmVP process can be easily adjusted to increase the resolution of the printed channels. In a pure TVAM process on the other hand, the pixel size would have to be reduced significantly to increase the resolution. This requires not only additional optics but also reduces the build volume.

We have made the following changes to the manuscript:

Direct TVAM printing of negative features allows to produce microchannels through a rapid process, but typically requires careful tuning of the projection patterns, especially with regards to diffusion effects. In our TVAM setup, the minimal channel diameter achievable is in the range of 500 μm , even for simple, straight 4 mm long designs, as shown Figure S4. To make the generation of smaller channels more accessible, we developed a complementary process to the positive EmVP additive ETVAM process presented above, is the negative EmVP, subtractive ETVAM process shown in Figure 1g,h and Figure 3.

To determine the channel size, we executed microCT scans of the printed parts and extracted the channel cross sections. The minimal channel diameter obtained through negative EmVP using a 150 μm diameter needle was 119 μm , shown in Figure S5. This is a definite improvement over pure TVAM in our setup, with which at 500 μm diameter only simple channels can be produced (see Figure S4).

We have added the following figures to the supporting information:

Figure S4. Cylinder structure with 4 mm long straight channels of different diameters to test negative feature reproduction for volumetric printing. (a) 3D model of the test structure with diameters of vertical

channels. Values are shown in mm. (b) Volumetrically printed sample showing clear overcuring starting from 0.5 mm diameter and fully clogged channels at 0.3 mm diameter. Scale bar (b): 2.2 mm

Figure S5. Model and cross section of a cylindrical structure with integrated microchannels to showcase negative EmVP. (a) Model of the cylinder printed by TVAM and the helix and straight channels deposited by EMB3D, with modeled diameter of 120 μm and 300 μm respectively. (b) Cross sectional view along the XY plane of the printed part, showing a channel diameter of 335 μm for the straight channel. Inset shows the cross-sectional view along the YZ plane of the helical channel, with a diameter of 119 μm. The higher diameter for the vertical channel is due to the print path requiring the needle to move through the same spot twice. Channels embedded using a needle with a diameter of 150 μm. Scale bar (b): 2 mm

Reviewer #2:

This manuscript from Tisato and colleagues about embedded tomographic additive manufacturing (ETVAM) describes a combined approach to 3D printing multimaterial structures by using tomographic VAM and embedded extrusion printing into a support bath. This is a useful addition to the growing range of capabilities for the tomographic volumetric AM paradigm. The authors demonstrate two modes, which they term additive ETVAM (in which the extrusion needle deposits a second material that cures to become part of the final structure), and subtractive ETVAM (in which the extrusion material is sacrificial and flushed away after final printing).

Overall, the work is well-executed, and the results are mostly clearly presented and described. However, for a journal with the broad audience of Nature Communications, I would hope for some more generality to be presented to readers, or some broader design principles beyond this initial demonstration. So I would recommend acceptance and publication of this work, but only after substantial revisions to answer at least some of the following questions:

Answer: We thank the reviewer for the positive assessment of our work.

Question 2.1a: *How does the resolution and accuracy of this approach, which combines EMB3D with TVAM, compare to the resolution and accuracy of each technique on its own?*

Answer 2.1a: We have added additional data to answer the question of the reviewer. The resolution of the TVAM printer is mainly governed by the resolution of the projected patterns and the oxygen

diffusion. Our custom printing system has a pixel size of 50 μm at the LCD, with which we have been able to print structures with positive features in the 300 μm range. To show this, we have included Figure S6 showing prints of different sizes.

For the EMB3D printer the resolution is chiefly a question of dimensions of the extruded filament and rheology of the ink and support bath. The EMB3D printer has a mechanical resolution of 5 μm for the motion system, and we were able to deposit ink filaments with a minimum diameter of $\sim 175 \mu\text{m}$ when depositing Mat 1 in Mat 2 at 15 mm/s. We have included Figure S7 to show the size of a single printed EMB3D filament.

With the combined EmVP process (please see answer 1.3 and 2.4 – in line with the request of the reviewer we have renamed our processes to "positive embedded extrusion-volumetric printing" (positive EmVP), replacing "additive ETVAM", and "negative embedded extrusion-volumetric printing" (negative EmVP), replacing "subtractive ETVAM") described in our work, the resolution of the EMB3D step is important to ensure accurate deposition of the embedded ink in positive EmVP and accurate deposition of the sacrificial ink in negative EmVP. Inaccurate deposition causes inaccuracies in the dose distribution being deposited during the TVAM step leading to distortion of the final prints in positive EmVP. For negative EmVP, it affects the channel shape and diameter. In positive EmVP the shape of the multi-material part is dictated by TVAM thus the resolution of the positive EmVP process is similar to the resolution of the TVAM step, which is in the range of 400 μm . The Hausdorff distance between 3D model and scan of print has been previously used to estimate the accuracy of prints (e.g. by Ribezzi *et al.*, [10.1002/adma.202409355](https://doi.org/10.1002/adma.202409355)). The positive EmVP process has an average accuracy of 0.23 ± 0.28 mm, as shown by Figure S8. The data shows that as the materials of the print gradually switch from fully Mat 1 to fully Mat 2, there are no significant changes in the Hausdorff distance. We therefore conclude that the accuracy of the positive EmVP process is similar to that of the TVAM process. Also, in this work, the created positive structures are defined by the TVAM step – the embedded print merely ensures the deposition of the material. Thus, after the TVAM curing there is remaining uncured embedded material in the vial (otherwise, the process would qualify as overprinting). To emphasize this we have added Figure S1, showing that a "shell" of uncured embedded material remains in the vial after the TVAM step. This shows that volumetric printing is carried out "inside" the embedded ink. Therefore, the accuracy of the embedded print has a limited effect on the accuracy of the overall EmVP process.

In the negative EmVP process on the other hand, the use of a non-polymerizing embedded ink to produce negative features allows us to ignore oxygen diffusion when generating the projections, reducing the computational cost. Here, the resolution of the EMB3D step is crucial for the definition of the channels, while the TVAM step is only responsible for the definition of the outer chip shape. By further tuning the embedding process, we were able to reduce the minimum channel size produced by negative EmVP to 119 μm , see Figure S5.

We have made the following additions to the manuscript:

The ~~positive EmVP additive ETVAM~~ process can produce stacked multi-material structures, by filling the print volume with layers of different material, as shown in Figure 2b,c, where a simple skeleton sphere structure is printed half in Mat 1 (bottom side) and half in Mat 2 (top side). Note that the embedded Mat 1 does not need to closely follow the shape of the final structure thanks to curing only selected areas inside the deposited Mat 1. This allows us to reduce the impact of common EMB3D issues such as unwanted oozing of ink during travel moves, shown in Figure S7a,b, by only polymerizing selected regions of the deposited ink, as shown in Figure S1. A close up of the structure in Figure 2b is included in Figure 2d, showing the transition between the two materials.

A magnified view is shown in Figure 2h highlighting the transition between the different materials. By calculating the Hausdorff distance, i.e. the average distance between the surface of the printed multi-material structure and that of the 3D model, shown in Figure S8, we obtain an average accuracy for the positive EmVP process of 0.23 ± 0.28 mm.

An overview of the total print times and the duration of the separate EMB3D and TVAM sections of the process is provided in **Table S1**. The characterization of the minimum printable positive features through our TVAM setup was carried out by printing increasingly smaller structures, shown in Figure S6, showing a minimum feature in the 300 μm range. For the EMB3D step, the diameter of the deposited liquid ink thread or “filament” was 175 μm , shown in Figure S7c,d.

Hausdorff distance. The Hausdorff distance was obtained by using CloudCompare “Cloud-To-Mesh Distance” function between the 3D model of the structure and the surface of the printed structure, extracted from a CT scan with 3D Slicer.⁴⁵⁻⁴⁷

45. CloudCompare - Open Source project. <https://www.cloudcompare.org/>. (2025)

46. Fedorov, A. et al. 3D Slicer as an image computing platform for the Quantitative Imaging Network. *Magnetic Resonance Imaging* 30, 1323–1341 (2012).

47. 3D Slicer image computing platform. 3D Slicer <https://slicer.org/>.

We have made the following addition to the supporting information:

Figure S6. Printing tests to assess resolution of the TVAM printer, showing decreasing sizes until first failures. (a) The printed structures, with external diameter of 10 mm, 5 mm and 2.5 mm respectively. (b) Microscope image of the beam for the 10 mm structure of (a), showing diameter of approx. 2 mm. (c) Microscope image of the beam for the 5 mm structure of (a), showing diameter of approx. 700 μm . (d) Microscope image of the beam for the 2.5 mm structure of (a), showing diameter of approx. 300 μm . At 2.5 mm external diameter obtaining an evenly resolved structure was not possible, as indicated by the uneven thickness of the spheres' truss. Scale bars: (a) 5.5 mm, (b,c,d): 650 μm

Figure S7. EMB3D printing of a spherical cage structure and deposition of a single ink filament to demonstrate the resolution of the EMB3D printer. (a) Sliced model of the structure, showing travel lines in blue between the beams. (b) Deposited Mat 1 in a Mat 2 support bath according to the Gcode generated from the slicing in (a). Note visible layer lines and the presence of unintended connection between beams due to the inability of the EMB3D printer to completely stop material flow during travel movements. (c) Microscopy image of a printed filament, with height of 250 μm . (d) Microscopy image of a cross section of a printed filament, showing rounded shape with diameter of approx. 175 μm ,

corresponding to the minimum deposited filament diameter printable with our system for this ink/bath combination. Scale bars: a 4.2 mm, b 11 mm, c 500 μm , d 250 μm .

Figure S8. Visualization of the printing accuracy for the multi-material structure of Figure 2g obtained by calculating the distance between the 3D model and the resulting part. (a) Printed positive EmVP structure. (b) CT scan of the structure of (a) colorized by the distance between the intended 3D model and a CT scan of the printed multi-material structure. Blue color indicates a match between the surface of the model and the print, while green yellow and red mark an increasingly high absolute distance of up to 1 mm (red). The data shows an average distance of 0.23 ± 0.28 mm between model and print. The Mat 1 region is contained in the dashed lines (c) Frontal view of the CT scan of (b) with Mat 1 region contained in the dashed lines. Along the Z direction there is a gradual change from Mat 2 to Mat 1. The data shows that there is comparable deviation between model and print for both areas (Mat1 and Mat 2), which suggests a similar accuracy for the presented EmVP process when compared to classical single-material VAM. Scale bars: 4.3 mm

Figure S1. Print process for positive EmVP and Comparison of prints in Mat 1, Mat 2 and combination of the two shown on prints of a spherical cage structure and showing proof that this approach is distinguished from EmVP overprinting. (a) Deposited Mat 1 in Mat 2 before curing. Note how the area of the deposited disk is fully filled. (b) Shadowgram of the positive EmVP of the spherical cage structure during the TVAM step. (c) Cured structure spanning across material boundaries. Note how the diameter of the printed structure is lower than that of the deposited disk of Mat 1. Thus, distinguishing this method

from previously shown overprinting approaches. (d) After the structure is removed there is leftover Mat 1 in the vial, further proving that only the desired areas of Mat 1 are cured and that Mat 1 is not fully immersed in Mat 2 after the print and hence this approach is not characterized as overprinting. (e) Printed multi-material structure extracted from the vial. (f) Visual comparison of structures printed by TVAM in a single material vial of Mat 2 (left), Mat 1 (center) and a combination of the two (right). Scale bars: a,d 12 mm, b 5.5 mm, c 10.5 mm, e 6 mm, f 7.5 mm

Figure S5. Model and cross section of a cylindrical structure with integrated microchannels to showcase negative EmVP. (a) Model of the cylinder printed by TVAM and the helix and straight channels deposited by EMB3D, with modeled diameter of 120 μm and 300 μm respectively. (b) Cross sectional view along the XY plane of the printed part, showing a channel diameter of 335 μm for the straight channel. Inset shows the cross-sectional view along the YZ plane of the helical channel, with a diameter of 119 μm . The higher diameter for the vertical channel is due to the print path requiring the needle to move through the same spot twice. Scale bar (b): 2 mm

Question 2.1b. *Are there limitations on geometric shapes/flexibility (one would imagine that many limitations of extrusion-based embedded printing remain, but only for one of the materials)? While a truly comprehensive study is probably too large in scope for an initial paper, the authors likely understand the overall trends to provide some comments and discussion for a reader interested in using this technique, but perhaps not with identical materials and formulations or printing conditions.*

Answer 2.1b: The reviewer is correct: some of the limitations of extrusion-based techniques remain in EmVP. One of the main limitations that remain is with oozing of ink in unwanted areas due to pressure buildup. This is a typical issue for EMB3D and can cause ink to be deposited in areas that should be printed exclusively in the support bath material, as now shown in Figure S7a-b, leading to undesired embedded ink curing in positive EmVP or voids in negative EmVP. This can be reduced with more complex extrusion setups, adjustment of the rheological properties of the ink or by designing the print path to avoid travel through specific regions of the vial. Using TVAM to cure only selected areas, and designing travel lines to fall in low dose regions, polymerization of unwanted travel lines can be avoided.

We have updated the discussion to include comments on the benefits and challenges coming from combining EMB3D and VAM, to make this clearer for the reader. Please see Answer 1.2 for details on the changes to the discussion section.

The following additions were made to the supporting information:

Figure S7. EMB3D printing of a spherical cage structure and deposition of a single ink filament to demonstrate the resolution of the EMB3D printer. (a) Sliced model of the structure, showing travel lines in blue between the beams. (b) Deposited Mat 1 in a Mat 2 support bath according to the Gcode generated from the slicing in (a). Note visible layer lines and the presence of unintended connection between beams due to the inability of the EMB3D printer to completely stop material flow during travel movements. (c) Microscopy image of a printed filament, with height of 250 μm . (d) Microscopy image of a cross section of a printed filament, showing rounded shape with diameter of approx. 175 μm , corresponding to the minimum deposited filament diameter printable with our system for this ink/bath combination. Scale bars: a 4.2 mm, b 11 mm, c 500 μm , d 250 μm .

Question 2.1c: Are there any principles that can be described, such as sources of distortion, feature blurring/minimum spacing? In Figure 3, for instance, the resulting channels are clearly distorted from a cylindrical/circular cross-section. How does such distortion compare to other embedded extrusion technologies?

Answer 2.1c: The main sources of distortion in positive EmVP are differences in the optical properties of the materials, such as refractive index and absorbance, misalignments when moving the vial between the steps of the process and inaccuracies in the extrusion process leading to inaccurate calculation of the projection patterns. For negative EmVP the rheological properties play an especially important role, as changes in yield stress of the bath can lead to a distortion of the printed line or “filament”, causing the ideally round cross-section of the filament to be distorted to oval or tear-drop shapes ([10.1002/admt.202400533](https://doi.org/10.1002/admt.202400533)), a typical distortion in EMB3D. While in positive EmVP this can be compensated for by depositing a slightly expanded version of the desired structure by embedded printing, for channels this is not as easily corrected.

The following general information is important when setting up an EmVP workflow:

- VAM is generally faster than material extrusion techniques (see for example [10.1126/sciadv.aao5496](https://doi.org/10.1126/sciadv.aao5496), Figure 4). Thus, to lower the print times, the embedded ink should be chosen for the part of the print that has a lower volume as this minimizes print time.
- Differences in absorption between the materials are likely to occur when designing new materials for positive EmVP, for example in our study Mat 1 has an absorption coefficient of 1.23 cm^{-1} while Mat 2 0.27 cm^{-1} . In this case it is beneficial to use the material with higher absorption as the embedded one, in our case Mat 1. Having a lower amount of higher absorption material allows more light to go through the vial, reducing print time of the VAM step.
- For materials with mismatches in gelation time, the difference in cure times causes the faster material to overcure. If the mismatch cannot be completely adjusted by changing the projections or the initiator content, the choice between whether to let the embedded or support material overcure depends on how the materials are placed in the structure to be printed and how well the EMB3D step can reproduce the intended volume. To improve shape

accuracy, letting the embedded material overcure can be beneficial as once the embedded material is cured, TVAM needs to cure the surrounding, slower curing support bath.

- For negative EmVP, the separation between channel geometry, deposited by EMB3D, and chip shape, cured by TVAM, reduces computational cost of the projection generation and simplifies the TVAM printing step, as it doesn't need to consider the channel and its overcuring. It also allows fast iteration as the same projection set can be used to produce chips with different path and diameter, as those are defined during EMB3D.

We have made the following addition to the discussion section to better convey this to the reader:

When setting up an EmVP process, some points need to be considered: we found that it is beneficial to have the embedded ink represent the part of the print with the lower volume as well as having the higher absorption. This allows to minimize the duration of the EMB3D step, as well as improve the TVAM step given the limited extent of the higher-absorbing material. We also note that when having a mismatch in cure time, the faster curing material being the embedded one helps obtaining prints with acceptable surface error, as once the embedded ink is polymerized, overcuring of the surrounding support bath starts when the remaining areas of the structure polymerize.

Please refer to Answer 1.2 for further discussion of the positive and negative EmVP processes.

Question 2.2: *Relatedly, the authors mention that after EMB3D printing, "the vial is moved to the VAM printer taking care to maintain alignment between reference frames." How is this alignment done? If by hand, what are the limitations on accuracy of alignment? If not by hand, please describe this.*

Answer 2.2: The alignment comes from the construction of the volumetric and embedded printers. The distance between the bottom of the vial and the lowest projection pixel is measured and used to set the origin of the EMB3D printer reference frame to the center of the vial in X and Y and the same distance in Z. The only alignment that need to be kept when moving between printers is the angular orientation, which in our case was checked by marking a point on the vial and aligning it to the same point on volumetric printer holder. We have found this to work well, but more complex system such as adding a non-curing projection wavelength would be needed when attempting to align smaller structures. With "alignment between reference frames" we mean that the positive directions of the X, Y, Z axes as well as the direction of rotation around Z have to match between the two sides of the printer. While this might sound obvious, due to the number of steps involved (model design, emb3d printing, projection generation, volumetric printing) each with its own reference frame, it is easy to introduce subtle, hard to debug errors.

We have added an additional schematic to the supporting information as Figure S9:

Figure S9 Visualization of the origins of the reference frames used at different steps of the process. (a) Reference frame for the 3D model. (b) Reference frame for the projection pattern (c) Reference frame of the EMB3D printer (d) Reference frame of the TVAM printer. All reference frames are set to have the origin at the center of the 3D model in the XY plane and at the bottom of the model in the Z direction.

Question 2.3. For additive ETVAM the authors talk about "similar gel times" for the two materials. How closely must these be matched? Here we see a difference of ~8% in terms of exposure time, but what are the consequences of a greater mismatch and what is the threshold? Is it important to match gel *times* or absorbed volumetric optical dose? How do the absorbances of the resins compare for the 0.5 wt% BAPO and 0.25 wt% CQ?

Answer 2.3: While in the best case both gel times and absorbed volumetric optical dose are matched, the match of the gel times should have a higher priority, as two materials with the same absorbed optical dose don't necessarily reach gelation at the same time when printing. In our case, the materials used have a different number of acrylate groups, with Mat 1 being a diacrylate while Mat 2 is a monoacrylate. Even assuming the neat materials have the same absorbance and using the same initiator, the difference in reactivity between the two would change the polymerization times significantly. With matched absorbed optical doses, this would lead to the faster one overcuring by the time the slower one finishes printing. Thus, matching absorbed optical dose is not sufficient to perform multi-material volumetric printing with positive EmVP. For a photopolymerizable support bath with low absorbance and a limited extent of the embedded material with a higher absorbance, we find compensating for the difference in absorbance is less challenging than compensating for differences in gel times. In our case, compensating for absorbance differences is done by considering the spatially varying absorbance coefficient during the computation of the projections. The threshold for the mismatch depends on how much the faster curing structure is allowed to deviate from a "perfect" print and how high the resolution of the EMB3D printing step is. As touched upon in question 2.1, by printing the embedded part of the structure exactly, a faster curing embedded ink would likely not cause excessive overpolymerization beyond the boundary of the embedded ink. Conversely, when a low-resolution embedded print is chosen, for example to print faster by simplifying the print path and increasing the size of the deposited filament, a better matching of gel times is necessary to correctly

reproduce features through volumetric printing of both materials at the same time. Regarding absorbance of the resins, we updated Figure 4c to show absorption coefficient instead of transmittance. Mat 2 with 0.25 wt% CQ has a transmittance of approximately 76.3 %, an absorbance of 0.118 and an absorption coefficient of 0.271 cm⁻¹, while Mat 1 with 0.5 wt% BAPO has a transmittance of 29.2 %, an absorption of 0.535 and an absorption coefficient of 1.23 cm⁻¹. Please see Answer 3.2 for the updated figure and the changes to the manuscript.

There are also a few minor corrections that I would suggest:

Question 2.4. *I'm not a fan of the "additive" vs. "subtractive" terminology for ETVAM, because additive is already used as an adjective on "additive manufacturing" in contrast to subtractive manufacturing, which is milling, cutting, drilling, etc. Perhaps "positive ETVAM" and "negative ETVAM" might be less over-loaded terms?*

Answer 2.4: We agree with the reviewer that further overloading "additive" could lead to confusion. We have decided to update our terminology to "positive embedded extrusion-volumetric printing", replacing "additive ETVAM", and "negative embedded extrusion-volumetric printing", replacing "subtractive ETVAM". For further details on the change of process names, please refer to Answer 1.2.

Question 2.5. *How was the silica loading % chosen? Was any sort of parameter sweep carried out to determine an optimal value? What is, for example, the minimum loading required to have these materials behave as desired for ETVAM?*

Answer 2.5: To pick a suitable silica loading we considered supporting properties and the ease of obtaining a print volume free of air bubbles. We tested several concentrations by performing 3-Interval-Time-Tests (3ITT) and flow curves, from 2% to 12% and picked a loading of 8 wt% that shows a low yield stress of ~12 Pa and recovery time of 1.7 s. The low yield stress helps with bubble removal when transferring to the printing vial as well as with the roundness of the filament cross section, which together with low recovery time increases EMB3D print resolution ([10.1002/admt.202400533](https://doi.org/10.1002/admt.202400533)). With a lower silica loading of 4 wt% and a yield stress of ~4 Pa, the material tends to flow and would not correctly retain the position of the liquid embedded ink when transferring between printers, while the higher loading of 12 wt%, transferring the prepared support bath to the print vial inevitably resulted in bubbles that cannot be removed without significant amounts of effort. The higher yield stress would also likely cause crevasse formation during EMB3D printing ([10.1002/admt.202400533](https://doi.org/10.1002/admt.202400533), [10.1021/acsami.2c08047](https://doi.org/10.1021/acsami.2c08047)). We have included the flow curves plots for viscosity and shear stress used to estimate yield stress of the tested loadings in the Supporting Information as Figure S2.

The following additions were made to the supporting information:

Figure S2 Rheological characterization of the support bath for different R805 loadings. (a) Flow curves showing increased viscosity with increasing loading, all showing clear shear thinning behavior (b) Flow curves showing increasing yield stress for increasing R805 loading.

Question 2.6. It would be excellent if the authors could at least estimate the Young's modulus values for Mat 1 and Mat 2, in addition to the Shore D durometer values, as the latter are difficult to compare to mechanical properties of other materials reported in the literature (durometer is significantly less frequently reported).

Answer 2.6: We have added the tensile tests for samples cast from Mat 1 and Mat 2 to the supporting information as Figure S10, showing the different tensile behavior of the materials. From the stress-strain curves, we estimate the elastic modulus for Mat 1 to be 122 MPa and for Mat 2 to be 1.28 MPa. We have updated relevant sections of the manuscript to include information about the elastic modulus.

The following subsection was added to the Methods section:

Tensile Testing. The tensile properties of cast samples of Mat 1 and Mat 2 were determined using an universal testing machine Inspekt Table (Hegewald & Peschke GmbH, Germany) following a scaled down ISO 527-1 norm.

The following addition was made to the supporting information:

Figure S10. Tensile test for cast sample of Mat 1 and Mat 2, showing very different mechanical properties. The elastic modulus is estimated to be 122 MPa for Mat 1 and 1.28 MPa for Mat 2.

The following changes were made to the manuscript:

Here we show the formation of parts with soft and hard properties, with the soft material having an elastic modulus of 1.28 and shore hardness of 4.9 and the hard material having an elastic modulus of 122 MPa and shore hardness of 27.4 (see Figure 1e and Figure S10).

The first one is a diacrylate (HDDA), characterized by a high elastic modulus of 122 MPa and a shore D hardness of 27.4. The second one is an aliphatic urethane acrylate, Genomer 1122TF (G1122TF), with an elastic modulus of 1.28 MPa and a shore D hardness of 4.9.

Question 2.7a. What is "SH resin" mentioned on Line 271? What is PLU mentioned on Line 293? Also, Pluronic does not have an "s" at the end of its name.

Question 2.7b. On line 281, the authors make reference to "product-derivative histogram equalization" from Ref 39, but that's incorrect (it's Proportional-Integral). Please check how Webber et al describe PIHE. Similarly, when mentioning PIHE on line 313, please cite ref 39.

Answer 2.7: We thank the reviewer for the thorough review and catching these mistakes that slipped through our writing process. SH resin was our initial name for "Mat 1", PLU was a shorthand name for Pluronic PE 3100. We adjusted the naming to be consistent with the rest of the manuscript and have added the requested citation and corrected the name from the optimization algorithm of Webber et al.

The followings changes were made to the manuscript:

Pluronic PE3100

Additionally, 0.015 wt% of Rhodamine B was added to Mat 1 the SH resin

while the embedding of **Pluronic PE3100 PLU** for the generation of microchannels

The projections for VAM are generated following a modified previously reported process and optimized by Object Space Model Optimization (OSMO) modified according to the **Proportional-Integral Product-Derivative** Histogram Equalization technique.^{38,39}

Question 2.8. *When observing whether a shadowgram for the VAM print, as described in lines 297-298 "matches with the desired shape," how is this measured? Using a camera? By comparison to an image/cross-section of the input model?*

Answer 2.8: The print vial is illuminated with a red LED from the side, orthogonally to the propagation direction of the projections. The shadow cast from the vial and printed part as it progresses is cast on a white screen. As this is similar to an optical transmission tomography projection of the printed part, the user can compare it to the projection used for the print process to determine the stopping point. We have added further clarification to the manuscript and an additional schematic of the print setup for the volumetric printer to the supporting information as Figure S11.

The following additions were made to the supporting information:

Figure S11 Schematic representation of the TVAM print setup. (a) A red LED (R) casts a shadow of the vial (V) on a screen (S) orthogonally to the projection (P) propagation direction, with which the user determines print completion. (b) A projected pattern (c) Corresponding red-light shadowgram at print completion.

Reviewer #3:

The authors show the combination of embedded extrusion and tomographic VAM to jointly cure two materials to produce 3D prints with selective material properties in the volume.

The authors show many examples, which are well realized and interesting.

Answer: We thank the reviewer for the positive assessment of our work.

The idea of using embedded 3D printing and VAM has been demonstrated previously in the context of bioprinting by Ribezzi et al “ Shaping Synthetic Multicellular and Complex Multimaterial Tissues via Embedded Extrusion-Volumetric Printing of Microgels” Adv. Mat. 2023. The study describes the use of embedded printing and VAM to produce multi-material constructs.

This study is referenced by the authors but it should be cited separately and the authors should explicitly highlight the difference with their work. In my opinion, the difference that the authors bring is that the embedded material is also polymerizable with the same gelation time as the matrix material and that they are both shaped by VAM resulting in a 3D print with the two material having very different mechanical properties.

Answer: The reviewer is right, Ribezzi *et al.* have introduced the concept of combining embedded printing and TVAM for biomaterials (EmVP). However, their work shows overprinting, the printing of one material around another material. There are no exposed areas of individual material properties obtained through EmVP. We have answered this question in more detail and explained the changes made to the manuscript in Answer 1.1. We have also added Figure S7 to better show how in our method the volumetric printing happens concurrently in both materials:

Figure S1. Print process for positive EmVP and Comparison of prints in Mat 1, Mat 2 and combination of the two shown on prints of a spherical cage structure and showing proof that this approach is distinguished from EmVP overprinting. (a) Deposited Mat 1 in Mat 2 before curing. Note how the area of the deposited disk is fully filled. (b) Shadowgram of the positive EmVP of the spherical cage structure

during the TVAM step. (c) Cured structure spanning across material boundaries. Note how the diameter of the printed structure is lower than that of the deposited disk of Mat 1. Thus, distinguishing this method from previously shown overprinting approaches. (d) After the structure is removed there is leftover Mat 1 in the vial, further proving that only the desired areas of Mat 1 are cured and that Mat 1 is not fully immersed in Mat 2 after the print and hence this approach is not characterized as overprinting. (e) Printed multi-material structure extracted from the vial. (f) Visual comparison of structures printed by TVAM in a single material vial of Mat 2 (left), Mat 1 (center) and a combination of the two (right). Scale bars: a,d 12 mm, b 5.5 mm, c 10.5 mm, e 6 mm, f 7.5 mm

Question 3.1: From the examples given in additive ETVM, it seems that all of them could have been realized with two material overprinting. The latter likely with more flexibility. Can the author describe what is the real advantage of their method ?

Answer 3.1: Two material overprinting as shown by Ribezzi *et al.* and Chansoria *et al.* is a vat exchange method, where one projection set solidifies the first material, then the uncured material is removed and the part is cleaned. A second material is then introduced into the vial containing the previously printed partial structure. After alignment, a second structure is cured using a second set of projections.

Challenges of this method are:

- vat-exchange and alignment both of the vial and of the partial structure. Aligning the partial structure in the vial to the projections is challenging and requires a modified printer with multiple projection wavelengths using a wavelength orthogonal to the curing wavelength for the alignment step. It also requires supporting structures to help maintain positioning of the structure printed in the first step. Without those, the structure would be free floating in the vial. While a convenient solution to the positioning problem, the use of supports eliminates one distinct advantage of TVAM, requiring post processing of the structure to remove supports like in traditional 3D printing. The alignment in the vial is further complicated by the fact that structures printed by TVAM are typically soft as they are just above the gelation threshold.
- bonding of the two materials besides intertwining or overprinting. To join two individual materials together with covalent bonds, the prints should not be postcured - as that would reduce the reactive groups on the surface available to bind to the second material. This means that the alignment has to be performed, likely manually, on soft structures placed inside the secondary material – it is difficult to immerse a soft structure in a viscous material, restricting the use of this technology to low-viscosity inks
- two sets of projections need to be designed, increasing computational cost. The second set of projections should also account for differences in refractive index and absorption of the first part of the structure to avoid compromising accuracy.

Example from [10.1002/adma.202409355](https://doi.org/10.1002/adma.202409355), Ribezzi *et al.*, Figure 4, showing support structures in SMVP (sequential multi-material volumetric printing):

In contrast to this, the presented positive EmVP (please see answer 1.3 and 2.4 – in line with the request of the reviewer we have renamed our processes to "positive embedded extrusion-volumetric printing" (positive EmVP), replacing "additive ETVAM", and "negative embedded extrusion-volumetric printing" (negative EmVP), replacing "subtractive ETVAM"):

- does not require alignment processes of the partial structure in the vial to match the alignment of the projections and does not require support structures. We only require to keep the same vial orientation in the two printers and to know the distance of the bottom projection pixel from the bottom of the vial along Z. When that is known, the EMB3D printer can simply be set to consider the origin of its coordinate system at that Z position in the center of the vial, and as long as the print vial is not rotated during the transfer between embedded and volumetric printer, the angular orientation is maintained. When compared to two material overprinting, this is an extremely simplified process which does not require additional locating structures to be printed and then removed or additional projection wavelengths to be built into the volumetric printer.
- only requires one set of projections, which significantly lowers computational cost

Please refer to Answer 1.1 and 1.2 for additional comments on the different existing methods for multi-material TVAM and the significant updates made to the introduction and discussion sections of the manuscript.

Question 3.2: *The light absorbance of the two materials is widely different, 30% vs 80% in figure 4 c. Is it measured for the same material length. It would be more indicative of giving the absorption of the material + Photoinitiator in units of 1/cm.*

Answer 3.1: The transmittance in Figure 4c is measured for the same cuvette with 1 cm optical path length. We have updated Figure 4c to show absorption coefficient in units of cm^{-1} instead of transmittance and updated the manuscript to match.

The following changes were made to the manuscript:

The **absorption coefficient transmission** at the working wavelength of 450 nm

The resulting **absorption coefficient transmission** curves are shown in Figure 4c

Figure 4. Photoreheological, rheological and optical characterization of the supporting baths and inks used in this work. (a) Photoreheological test of Mat 1, showing gelation being reached at 59.8 s from the start of the measurement. (b) Same test as in (a) for Mat 2, showing similar time to gelation of 64.7 s. As the time to gelation are similar, both materials cure at similar times when undergoing VAM together. (c) Absorption coefficient Transmittance of the two materials, highlighting values at the VAM printer LED emission wavelength. The low absorbance – high transmittance of Mat 2 makes it suitable for VAM, while Mat 1 would normally have too high absorbance little transmission if used directly for VAM, but thanks to being only embedded in specific regions of the volume, it can still be used. (d) Stress/recovery Thixotropic time test of Mat 1, showing a single high-low shear cycle. The recovery time for Mat 1 is 4.7 s. (e) Same test as in (d) for Mat 2, showing a low recovery thixotropic time of 1.7 s, making it suitable as a support bath for EMB3D. (f) Flow curves of Mat 1 and Mat 2, showing shear thinning behavior for both. For Mat 1, this aids in obtaining a smooth extrusion flow during EMB3D.

Question 3.3: It is not clear if the pattern computation takes into account the light absorption of the combined material. In other words, it seems that the 3D location of the first material needs to be known in the print in order to account for the propagation of light which can traverse both material.

Is the index of refraction difference between the two material taken into account in the computation of the pattern? A small difference would make the rays deviate from a straightline. This would compromise resolution.

Answer 3.3: The pattern computation does take the absorption coefficient into account by using the appropriate absorption coefficient for each resin. The reviewer is correct in that the 3D location of Mat 1 needs to be known to model the absorption, this is done by using a spatially varying absorption coefficient during projection computation, as mentioned in the section “Methods - Projection generation”. The difference in refractive index is not currently accounted for in the computational model. As correctly explained by the reviewer, variation of refractive index would deviate rays from the expected direction of propagation. As adding the index of refraction to the pattern computation would lead to a very significant increase in computational cost and complexity, we opted for minimizing this effect by choosing materials with similar refractive indices, as Mat 1 has a refractive index of 1.456, close to the refractive index of Mat 2 at 1.4598. We have found this to perform well for our TVAM setup. For even higher complexity prints that require high amounts of embedded ink, this would need to be considered.

We have added the following note to the manuscript to make this clearer for the reader:

In our case, an addition of 7 wt% of R805 to HDDA and 8 wt% of R805 in G1122TF provide suitable properties to act as an embeddable ink in the case of HDDA (Mat 1) and as a support bath in the case of G1122TF (Mat 2), both of which are suitable for VAM. Differences in refractive index of the materials deviate the light propagation direction from the assumed straight line. To reduce the effect of refractive index mismatch, we have chosen materials that show close refractive indices, as reported by the suppliers, with 1.4598 and 1.456 for Mat 1 and Mat 2, respectively. To demonstrate the capabilities of our process positive EmVP to produce complex, multi material structures, we have printed several structures as shown in **Figure 2**.

The projection generation, implemented in Python, is carried out on the bwUniCluster shared cluster running an Intel Xeon Gold 6230 and Nvidia Tesla V100. Forward, backward and optimization example code is available as **Code S1**.

We have added the code used for the projection calculation to the supporting information:

Code S1. Python code for projection computation

```
import torch
import torchvision.transforms.v2.functional as tv
from tqdm import tqdm

def cylinderMask(model):
    # mask cylinder matching the diameter of the model voxel volume

    circle_y, circle_x, circle_z = torch.meshgrid(
        torch.linspace(-1, 1, model.shape[0]),
        torch.linspace(-1, 1, model.shape[1]),
        torch.linspace(-1, 1, model.shape[2]),
        indexing="xy"
    )

    model[~((circle_x ** 2 + circle_y ** 2) <= 1).bool()] = 0
    return model

def model_to_sino(model, alpha, step=0.005, maxangle=360, dtheta=1, trunc=True,
byProj=False, interp=tv.InterpolationMode.BILINEAR):
    """
    Forward projection using space-varying abs coeff.

    model: XxYxZ voxels (X==Y)
    alpha: same shape as model, absorption coeff. in cm-1
    step: discretization of the model, in cm
    maxangle: maximum angle to generate projections for, deg
    dtheta: angular distance between projections, deg
    trunc: whether to truncate the sinogram at 0
    byProj: whether to scale each projection by its own max (true) or the max of the set of
    projections (false)

    returns sino as Y, Theta, Z
    """

    # model & alpha should both be (x, y, z)
    assert model.shape == alpha.shape, "model and alpha must match in shape"

    model_u = model.permute(2, 0, 1).contiguous() # (z, x, y)
    alpha_u = cylinderMask(alpha).permute(2, 0, 1).contiguous() # same shape

    zdim, _, ydim = model_u.shape
    N_angles = int(maxangle / dtheta)
    sino = torch.zeros((N_angles, zdim, ydim), device=alpha.device, dtype=alpha.dtype) #
    theta, z, y
```

```

for theta_idx in range(N_angles): # rotate model and calculate new projection
    theta = theta_idx * dtheta
    model_rot = tv.rotate(model_u, theta, interp)
    alpha_rot = tv.rotate(alpha_u, theta, interp)

    weights = alpha_rot * step * torch.exp(-torch.cumsum(alpha_rot * step, dim=1))
    proj = torch.sum(model_rot * weights, dim=1)
    sino[theta_idx, :, :] = proj

if trunc: sino.clamp_(min=0)

if byProj:
    max_per_proj = sino.amax(dim=(1, 2), keepdim=True)
    sino = sino / (max_per_proj)
else:
    sino /= sino.max()

return sino.permute(2, 0, 1) # (y, theta, z)

def sino_to_rec(sino, alpha, step, dtheta=1, interp=tv.InterpolationMode.BILINEAR):
    """
    Backward projection using space-varying abs coeff.

    sino: YxThetaZ voxels
    alpha: XxYxZ voxels (X==Y), absorption coeff. in cm-1
    step: discretization of the model, in cm
    dtheta: angular distance between projections, deg

    returns rec (XxYxZ)
    """
    alpha_u = cylinderMask(alpha).permute(2, 0, 1).contiguous() # from X Y Z to Z X Y
    sino_u = sino.permute(1,2,0).contiguous() # from y theta z to theta z y
    zdim, xdim, ydim = alpha_u.shape
    N_angles = sino.shape[1] # sino is y theta z
    rec = torch.zeros_like(alpha_u)
    absb = torch.zeros_like(rec)

    # compute contribution of projection theta to reconstruction:
    for theta_idx in range(N_angles):
        theta = theta_idx * dtheta
        # rotate alpha so that the propagation considers the spatially varying abs coeff
        alpha_rot = tv.rotate(alpha_u, theta, interp)
        I_proj = sino_u[theta_idx, :, :].unsqueeze(1).expand(alpha_rot.shape).contiguous() #
z,y to z,1,y to z,x,y
        for x in range(1,xdim):
            I_proj[:,x,:] = I_proj[:,x-1,:]*torch.exp(-alpha_rot[:,x,:]*step)
            absb[:,x-1,:] = (I_proj[:,x-1,:]-I_proj[:,x,:])
        # rotate contribution to match with angle theta and add up into rec
        rec += tv.rotate(absb, -theta, interp)
    return rec.permute(1,2,0)

def opt(model, sino, rec, niter, alphas, px=0.005, maxOutDose =.60, minInDose=.80, k_p=1.0,
k_i=0, maxangle=360, dtheta=1, byProj=True):
    model_n = torch.clone(model)
    # Initial guess
    sino_n = torch.clone(sino)
    rec_n = torch.clone(rec)

    voidInds = model<1 # out part
    gelInds = model>0 # in part

    threshold_L = maxOutDose
    threshold_H = minInDose
    E_sum = torch.zeros_like(rec_n)

```

```

for i in tqdm(range(niter), desc="pi-osmo"):
    rec_n /= torch.max(rec_n) # normalize reconstruction
    E = torch.zeros_like(rec_n)
    E[gelInds] = threshold_H-rec_n[gelInds]
    E[voidInds] = rec_n[voidInds]-threshold_L
    E[E < 0] = 0
    E_sum += E

    # update step
    model_n[voidInds] -= k_p*E[voidInds] + k_i*E_sum[voidInds]
    model_n[gelInds] += k_p*E[gelInds] + k_i*E_sum[gelInds]
    sino_n = model_to_sino(model_n, alphas, px, maxangle, dtheta, byProj=byProj)
    rec_n = sino_to_rec(sino_n, alphas, px, dtheta, byProj=byProj)

return sino_n, rec_n

#
if __name__ == "__main__":
    model = loadModel("test.stl") # load some model as a binary voxel volume, XxYxZ. X==Y
    alpha = 0.7 # cm-1
    step = 0.005 # cm, == pxsize
    sino = model_to_sino(model, alpha, step)
    rec = sino_to_rec(sino, alpha, step)
    s_opt, r_opt = opt(model, sino, rec, niter=100, px=step, maxOutDose= 0.82,
minInDose=0.92, k_p=1.0, k_i=0.5, byProj=True)

```

Question 3.4: The gelation time is measured from rheology, however it is known in VAM that diffusion tend to increase the gelation time for smaller structures. Given the two different material light absorption, it would seem difficult to make arbitrary 3D structures for both materials simultaneously. The authors did not comment on resolution and the ability to generate arbitrary structures.

Answer 3.4: It is indeed challenging to print arbitrary structures, but as seen in Figure 2, we could print a range of different parts, some with a mix of finer and bulkier features and having the deposited material occupy varying regions of the model, as shown in Figure 2f. It is also important to note that the embedded material with higher absorbance is only present in limited regions, which reduces the impact of the higher absorbing material. We added Figure S12 to the supporting information to illustrate how the inclusion of a higher absorbing Mat 1 in the vial results in higher intensity at the exit of the vial volume when compared to a vial fully filled with Mat 1 only.

Please see Answer 2.1 for additional comments on the resolution. In short, our print setup is able to reproduce structures with minimum features in the 400 μm range in Mat 2. The average Hausdorff distance between 3D model and CT scan was 0.23 ± 0.28 mm.

We added the following figure to the supporting information:

Figure S12. Simulated intensity through the print volume for a vial with addition of different amounts of Mat 1 in Mat 2. (a) 2 mm long Mat 1 inclusion (b) 6 mm long Mat 1 inclusion (c) 8 mm long Mat 1 inclusion. Note how the intensity at the exit of the vial for the combined Mat1 and Mat2 volume is higher than the one for a full Mat 1 volume and for lower Mat 1 inclusion volumes, approximates the intensity of a fully Mat 2 filled vial. Dotted lines represent vials fully filled with Mat 1 or fully filled with Mat 2.

Response to the reviewers' questions

Dear reviewers, dear editors,

we thank the reviewers for the helpful suggestions they have made. We have addressed all the suggestions and issues raised – please find our response to the comments of the reviewers on the following pages. The reviewer's comments are set in italic whereas our comments are set in upright font. Where relevant, manuscript excerpts in Times New Roman with changes highlighted in yellow have also been added here for clarity.

Reviewer #1 (Remarks to the Author):

The authors carefully revised the manuscript and addressed the previous concerns.

Answer: We thank the reviewer for the positive assessment of our work.

Reviewer #2 (Remarks to the Author):

I very much appreciate the authors' thorough response. In my opinion they have adequately addressed all the comments and suggestions made by the referees. I particularly appreciate the addition of Figure S8 to quantitatively assess the printing accuracy of the final structure in comparison to the source model. I recommend publication of the revised manuscript.

Answer: We thank the reviewer for the positive assessment of our work.

Reviewer #3 (Remarks to the Author):

From the reviewers questions, all have indicated that explaining better the difference between the proposed embedded extrusion and VAM with respect to other works demonstrating embedded extrusion is essential.

To me, this is important for the readers to understand which method to use when combining embedded extrusion with VAM. There are advantages and disadvantages. I believe a table highlighting the pros and cons would be very useful and clear a possible confusion

Answer: We have added a summary of the advantages and disadvantages of the different multi-material methods developed for VAM to the supplementary information as Supplementary Table 2 and added a note to the discussion.

The following addition was made to the discussion:

TVAM has been used to accurately reproduce simple, straight microchannels with reported diameters as low as 124 μm with the use of high-resolution volumetric printers while for more complex channels the achieved channel diameters in the range of 250-500 μm .^{23,29,39,40} A comparison of multi-material methods developed for TVAM showing advantages and disadvantages is summarized in Supplementary Table 2. In the case of the printer used in this work, diffusion effects and overcuring make printing negative features lower than 500 μm challenging, as shown in Figure S4.

The following table summarizing advantages and disadvantages of different methods for multi-material VAM was added to the supplementary information:

	EmVP	Vat exchange overprinting	Positive EmVP (this work)
Place different materials in the bulk of the part	Yes	Yes	Yes
Place different materials on the surface of the part	No	Yes	Yes
Matrix needs to be engineered to possess supporting properties	Yes	No	Yes
Requires at least one ink to be extrudable	Yes	No	Yes
Requires matched curing properties of the individual materials	No	No	Yes
Need to align structures in supporting material	No	Yes	No
Simultaneous shaping of multiple materials by TVAM	No	No	Yes
Single projection set	Yes	No (one set per material)	Yes
Requires EMB3D printer	Yes	No	Yes
Requires support structures to aid alignment	No	Yes	No
Requires multi-wavelength projector to perform alignment	No	Yes	No
Need to perform material exchange	No	Yes	No
High positioning freedom for secondary material	Yes (no surface)	No (only unoccupied areas)	Yes (surface and bulk)

Supplementary Table 2. Comparison of multi-material methods developed for TVAM, showing advantages and disadvantages.